# Faults in Our Formal Benchmarking: Dataset Defects and Evaluation Failures in Lean Theorem Proving

**Pawan Sasanka Ammanamanchi** [1]  **Siddharth Bhat** [2]  **Stella Biderman** [1]

## Abstract

Benchmarks for LLM-assisted theorem proving in Lean are often treated as intrinsically reliable because every solved instance comes with a machine-checked proof. However, the kernel only checks that a proof establishes a *formal* statement; it does not verify that the statement faithfully encodes the intended informal problem, nor that evaluation harnesses are robust to trivial or adversarial solutions. We audit five widely used Lean theorem-proving benchmarks and their forks, using corpus-scale static checkers to surface 4,833 findings, including 398 mechanically certified issues such as counterexamples, vacuous theorems, and unsound axioms. We also document semantic defects such as missing hypotheses, problem simplification, incomplete or incorrect translations, and Lean-specific specification hazards. Beyond dataset construction, we survey evaluation-time failure modes and show, on corrected subsets, that defects can both inflate and deflate reported prover scores. We propose a fault taxonomy, a suite of automated checkers and recall-oriented semantic-audit prompts, and release standards to guide the creation of formal math datasets and make evaluation more reproducible and trustworthy. Our checkers, audit prompts, and corrected dataset snapshots are available at `https://github.com/Shashi456/atp-checkers`.

## 1. Introduction

Deep learning has rapidly advanced automated theorem proving, with models increasingly trained to produce proofs in Lean. Recent systems such as DeepSeek-Prover V2 (Ren et al., 2025), Goedel Prover 2 (Lin et al., 2026), and Kimina Prover (Wang et al., 2025) report progress on widely used

benchmarks such as miniF2F and ProofNet. However, as in other LLM evaluation settings, benchmark scores can be misleading when problem statements are mis-specified or become outdated, or when evaluation protocols contain loopholes.

A common intuition is that Lean benchmarks are "self-verifying" because the kernel checks every proof. This intuition is incomplete. The Lean kernel provides certainty about a narrow claim: a given proof artifact establishes a given *formal statement*. This provides mathematical certainty about the formal claim, but overall benchmark reliability is limited by specification accuracy and by the correctness of the natural-language-to-Lean translation. Lean does not certify that the formal statement matches the intended informal problem, that the benchmark avoids Lean-specific semantic pitfalls, or that the evaluation protocol cannot be exploited. These gaps can inflate reported success without demonstrating stronger proof capability.

*Table 1.* Benchmarks referenced in this study (details in §3).

| Dataset | #Prob | Lean | Some Issues |
|---|---|---|---|
| miniF2F | 488 | 3 | Version proliferation, incomplete formalization, type errors |
| ProofNet | 371 | 3 | Underspecification, incomplete formalization |
| FormalMath | 5,560 | 4 | Arithmetic errors |
| CombiBench | 100 | 4 | Incomplete spec., incorrect translation |
| ProverBench | 325 | 4 | Wrong problems, incomplete spec. |

In this paper, we audit five widely used Lean benchmarks and their forks (Table 1). We analyze the benchmark development pipeline (Section 2) and develop a fault taxonomy (Section 3) that separates formalization-time fidelity failures, evaluation-time loopholes, and maintenance decay. To scale detection beyond manual review, we implement automated static checkers as Lean 4 metaprograms and run them corpus-wide across all 13 benchmark variants (~10,000 problems), surfacing 4,833 findings of which 398 carry a

[1]Eleuther AI  [2]Cambridge.  Correspondence to: Pawan <pawansasanka@gmail.com>.

*Proceedings of the 43rd International Conference on Machine Learning*, Seoul, South Korea. PMLR 306, 2026. Copyright 2026 by the author(s).

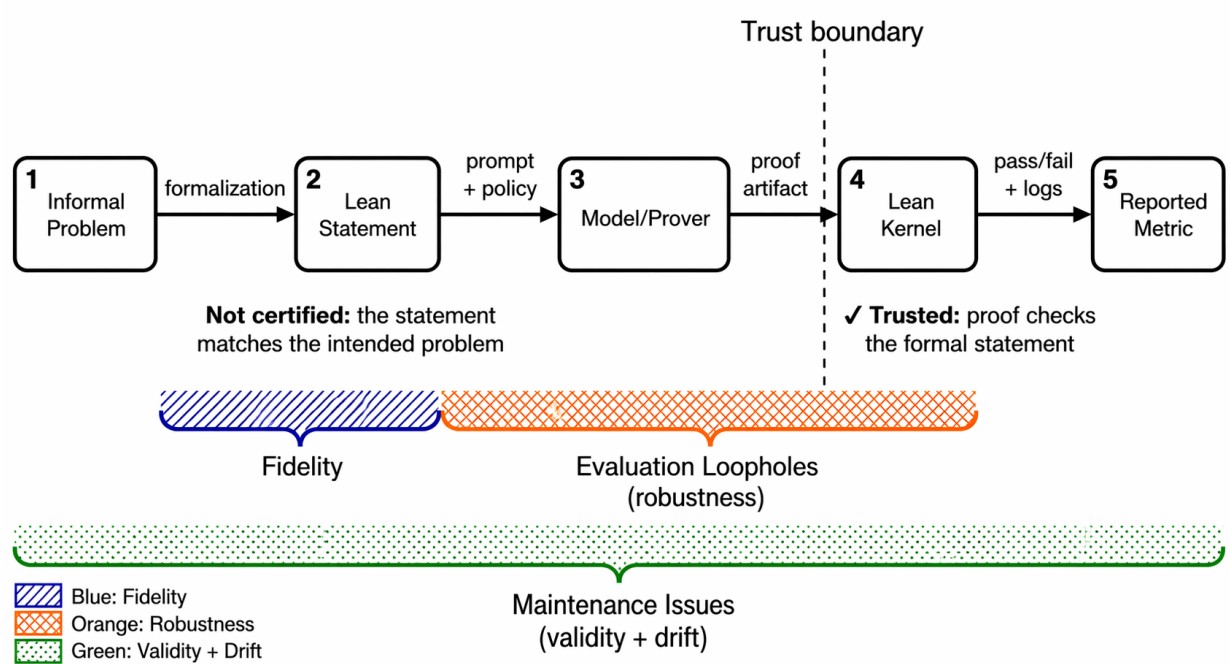

*Figure 1.* The formal benchmarking pipeline and where errors can enter. The kernel (trust boundary) certifies only that a proof artifact establishes the *formal* Lean statement. Three categories of issues arise: **Fidelity Issues** in the translation step, **Evaluation Loopholes** (robustness) in the proof-checking and reporting steps, and **Maintenance Issues** (validity + drift) spanning the entire pipeline. Table 2 details the specific fault types within each category.

machine-checkable certificate of unprovability or vacuity; we additionally evaluate LLM-based prompts on a labeled 92-problem challenge set for the semantic mismatches static analysis cannot decide, and measure how corrected statements change reported prover scores (Section 4). Finally, we propose release standards and tooling to harden future dataset creation and evaluation (Section 5).

## 2. What Formal Benchmarking Certifies (and What It Does Not)

Formal benchmarks are typically adapted from informal problems through a series of steps that can undermine the validity of the benchmark. Figure 1 illustrates this pipeline. We walk through each stage below, noting where errors can enter the pipeline.

**Step 1: Informal Problem.** The pipeline begins with an **informal problem**, an Olympiad question, textbook exercise, or research conjecture stated in natural language. Issues can already arise at this stage: the source material may contain errors, the problem may be ill-posed or ambiguous, the mathematical claim may be false. These are *validity* issues that no amount of careful formalization can fix.

**Step 2: Formalization.** The informal problem is then **formalized** into a Lean statement. This translation step is where most defects arise (*fidelity* issues). Hypotheses can be omitted, domains mistranslated (e.g., $\mathbb{N}$ instead of $\mathbb{Z}$), or Lean-specific encoding issues can silently change meaning or make a theorem vacuous. A proof of the resulting formal statement may be mathematically valid yet prove something different from what the informal problem intended.

**Step 3: Model/Prover.** A model or automated prover receives the Lean statement (plus a prompt and policy) and attempts to produce a **proof artifact**.

**Step 4: Lean Kernel.** The proof artifact is checked by the Lean kernel, which emits a pass/fail result. This is the **trust boundary**: the kernel provides mathematical certainty that the proof artifact establishes the formal statement. However, models can find bugs in the Lean environment thus producing proofs that appear to be "accepted" without proper kernel verification (*robustness* issues).

**Step 5: Reported Metric.** Finally, the evaluation harness aggregates pass/fail results into a reported metric. Loopholes at the previous stage can inflate scores without demonstrating genuine capability.

**What the kernel certifies.** The key insight is that Lean's kernel, the only component providing mathematical certainty, operates at a single point in this pipeline. The trust boundary in Figure 1 marks the limit of what Lean guarantees: everything to the left of the kernel and everything to the right must be validated by other means. It does *not* certify that the formal statement matches the intended informal problem (Step 2), nor that the informal problem itself is correct (Step 1), nor that the evaluation protocol and metric reporting are sound (Step 5).

We organize these failure modes into three categories (Figure 1, bottom; Table 2): **Fidelity Issues** arise in Step 2; **Evaluation Loopholes** arise in Steps 4–5; **Maintenance Issues** span the entire pipeline, including source errors in Step 1 and semantic drift as mathlib evolves over time.

## 3. Fault Taxonomy and Representative Failures

Our audit of miniF2F, ProofNet, FormalMath, CombiBench, and ProverBench revealed that benchmark defects cluster into categories requiring fundamentally different interventions. We organize faults by where they enter the benchmarking pipeline (Figure 1) and what fix they require (Table 2):

- **Fidelity issues**: issues during formalization require careful review at creation time/ lean encoding hazards can be detected by static analysis

- **Evaluation loopholes**: require stricter harnesses and patched Lean versions

- **Maintenance decay**: requires version pinning and active stewardship

One cannot "fix" a kernel bug/bypass by adding dataset review, nor prevent missing hypotheses by restricting tactics. The taxonomy makes remediation actionable by matching faults to appropriate interventions. We illustrate each category with representative defects from our audit and tag every example with its (sub)category; further examples appear in Appendix B.

### 3.1. Fidelity Issues

These faults occur when the Lean statement does not faithfully encode the intended informal problem. We distinguish two primary failure modes:. *Specification errors* omit required content, such as missing hypotheses, dropped subgoals, or unencoded constraints, leaving the statement valid but *weaker* than intended, and therefore easier to prove than the original. *Formalization errors* instead misrepresent the original, through incorrect translation, wrong quantifier scope, or encoding artifacts, so the statement may become

*unprovable* or prove *something different*. The two demand opposite repairs: a specification error is fixed by *adding* the missing content, a formalization error by *correcting* the encoding.

**Missing hypotheses.** *(Specification)* Forgetting properties of mathematical objects can make problems trivial, vacuous, or false. In Figure 2, the formalization omits the requirement that $V$ be finite-dimensional, a hypothesis explicitly stated in the informal problem and essential for the result.

---

**ProofNet – Axler Exercise 3.8**

**Problem:** Suppose that $V$ is **finite dimensional** and that $T \in \mathcal{L}(V, W)$. Prove that there exists a subspace $U$ of $V$ such that $U \cap \text{null } T = \{0\}$ and range $T = \{Tu : u \in U\}$.

- - - - - - - - - - - - - - - - - - - - - - - - - - - - - - - -

```
theorem exercise_3_8 {F V W : Type*} [add_comm_group V]
  [add_comm_group W] [field F] [module F V] [module F W]
  (L : V →ₗ[F] W) :
  ∃ U : submodule F V, U ⊓ L.ker = ⊥ ∧
    linear_map.range L = range (dom_restrict L U) := sorry
```

---

*Figure 2.* Missing hypothesis: the Lean statement omits finite-dimensionality, which is required for the theorem to hold.

**Incomplete translation.** *(Formalization)* Problems often have several parts, and a formalization can silently drop one. In Figure 3, the Lean statement captures the inequality but omits the "determine when equality occurs" requirement, encoding only half of the intended problem and leaving it strictly weaker than the original. In another example Figure 5, the formalization leaves $u$ as a free variable without specifying $u = y - 2x$.

---

**miniF2F – IMO 1983 Problem 6**

**Problem:** Let $a$, $b$ and $c$ be the lengths of the sides of a triangle. Prove that $a^2 b(a-b) + b^2 c(b-c) + c^2 a(c-a) \geq 0$. **Determine when equality occurs.**

- - - - - - - - - - - - - - - - - - - - - - - - - - - - - - - -

```
theorem imo_1983_p6 (a b c : ℝ) (h₀ : 0 < a ∧ 0 < b ∧ 0 < c
  )
  (h₁ : c < a + b) (h₂ : b < a + c) (h₃ : a < b + c) :
  0 ≤ a^2 * b * (a - b) + b^2 * c * (b - c) +
    c^2 * a * (c - a) := sorry
```

---

*Figure 3.* Incomplete specification: the equality-case determination is missing from the formalization.

**Wrong domain or type.** *(Domain & Definition mismatch)* Translating variable domains inaccurately changes problem semantics. If a problem states "for all positive integers $n$," using (n : ℕ) without a guard (hn : 0 < n) includes $n = 0$. Conversely, using ℕ when ℤ is intended can introduce truncation errors.

*Table 2.* Taxonomy of faults in Lean theorem proving benchmarks.

| Category | Subcategory | Issues |
|---|---|---|
| **Fidelity** | Specification | Missing hypotheses/conditions; extra or incorrect constraints; incomplete or over-simplified spec |
| | Formalization | Goal/premise translation errors; missing subgoals; vacuous hypotheses |
| | Domain & Definition | Wrong type; wrong mathlib concept; misuse of library API; encoding artifacts |
| | Lean Encoding | Nat subtraction truncation; division/mod by zero; Int/Nat coercion |
| **Evaluation** (robustness) | Proof Acceptance | Improper axiom usage; `native_decide` shortcuts; unsafe tactics |
| **Maintenance** (validity + drift) | Decay | Version drift (Lean 3→4); dependency rot (mathlib API changes) |
| | Benchmark Defects | NL statement defects; unprovable statements; harness drift |

**Nat subtraction and division hazards.** *(Lean Encoding)* Even when a statement "looks correct," Lean semantics can silently change meaning: We found multiple instances in FormalMath and ProverBench where $\mathbb{N}$ subtraction silently changed problem semantics (Appendix B).

**Definition mismatch.** *(Domain & Definition mismatch)* Sometimes the Lean encoding uses an incorrect or outdated mathlib concept. This is a mathematical representation issue.

**Vacuous hypotheses.** *(Formalization)* Incorrect translation can produce contradictory hypotheses, making statements vacuously true. For example, an earlier CombiBench formalization encoded a minimality condition as $\forall N' < N, \neg \text{Periodic}(W, N')$, but `Periodic W 0` is trivially provable in lean with `simp`. This made the hypotheses unsatisfiable, allowing a trivial `exfalso` proof (Appendix E).

### 3.2. Evaluation loopholes

Even if the benchmark statement is correct, an evaluation can be invalidated when the harness accepts a proof artifact that does not demonstrate the intended capability.

**The `apply?` frontend bug.** A bug in Lean versions prior to 4.20.0 allowed the `apply?` tactic to report success without producing a theorem declaration that had passed ordinary kernel verification: a synthetic sorry interacted with a universe-level mismatch and bypassed the usual "declaration uses sorry" warning. This failure mode appeared in at least three proofs claimed by DeepSeek-Prover-V2 (Ren et al., 2025): one miniF2F solution and two PutnamBench solutions from the 7B model (Appendix D). The bug has since been fixed,[1] but it illustrates a broader concern: *RL-trained*

*provers will find and exploit any verification loophole that increases reward*.

The Lean community has since released `comparator`,[2] a trusted evaluator that checks proofs against a conservative set of axioms and disallows potentially unsafe tactics like `native_decide`. While `comparator` does not address all end-to-end trustworthiness issues (e.g. the `apply?` bug, which operated below the tactic layer), it represents an important step toward trustworthy evaluation. We recommend that evaluation harnesses use patched Lean versions, verify `#print axioms` output, and incorporate maximally strict verification in RL reward signals.

**Native_decide trust expansion** `native_decide` expands the trusted computing base beyond the kernel by trusting the compiler/codegen result via the `Lean.ofReduceBool` axiom. This expands the trusted code base from the kernel to the entire compiler: known bugs in native code generation and implemented_by overrides have produced proofs of False. Evaluation harnesses should disallow `native_decide` or carefully restrict its use.

**Axiom injection.** If the environment contains `axiom h : P`, then any solver can "prove" $P$ by citing $h$. Benchmark harnesses should forbid axioms in problem files and in any imported modules.

**Over-permissive tactics and unsafe automation.** Tactics like `decide` can solve goals by computation when that is intended, but they can also collapse tasks when the goal is improperly typed or overly concrete. A robust harness should specify which tactics are allowed and record the exact proof artifact.

---

[1] https://github.com/leanprover/lean4/pull/8231

[2] https://github.com/leanprover/comparator

**Problem Statement**

**Rudin Ex. 2.28:** Prove that every closed set in a separable metric space is the union of a (possibly empty) perfect set and a set which is at most countable.

**ProofNet (Incomplete)**

```
theorem exercise_2_28 (X :
      Type*)
  [metric_space X] [
      separable_space X]
  (A : Set X) (hA :
      is_closed A) :
  ∃ P₁ P₂ : Set X, A = P₁
      ∪ P₂ ∧
  is_closed P₁ ∧
  P₁ = {x | cluster_pt x
      (ℙ P₁) } ∧ set.
      countable P₂ :=
    sorry
```

**ProofNet# (Correct)**

```
theorem exercise_2_28 (X :
      Type*)
  [MetricSpace X] [
      SeparableSpace X]
  (A : Set X) (hA :
      IsClosed A) :
  ∃ P₁ P₂ : set X, A = P₁
      ∪ P₂ ∧
  Perfect P₁ ∧
  Set.Countable P₂ :=
    sorry
```

*Figure 4.* Version and definition drift: the original ProofNet version encodes "perfect set" using closure and cluster points; the later version uses mathlib's `Perfect` predicate.

### 3.3. Maintenance & Source Issues

Even a faithful, well-evaluated benchmark degrades over time. Lean and mathlib evolve rapidly; benchmarks released against a specific snapshot can stop compiling within months, and quick fixes can introduce semantic drift. Some problems are defective at the source (ambiguous, ill-posed, or false NL statements). Others become defective through *harness drift*: inconsistent versioning updates and unrecorded benchmark changes, across evaluation setups, make reproducibility unreliable.

**Version drift.** *(Maintenance Decay)* The Lean theorem prover (de Moura et al., 2015) evolves rapidly, with new tactics and library conventions shipping in successive releases. Lean 4's first stable release was on September 8, 2023, and new versions arrive within months; a benchmark pinned to one Lean/mathlib snapshot can fail to compile as APIs change, or shift in meaning as definitions are revised. For example, when ProofNet was first released the mathlib predicate `Perfect` did not yet exist, so the authors encoded a perfect set as one that is closed (`IsClosed`) and equal to its set of cluster points (Figure 4). The later ProofNet# version uses mathlib's `Perfect` predicate, which exposes a much larger API and changes the nature of the mechanization. In paper mathematics the two encodings are interchangeable, and a mathematician would move freely between them; in a mechanized setting that flexibility is not free, and substituting one for the other requires first proving their equivalence.

A similar difference between paper-and-pen mathematics and formal mathematics is that of limits. Limits in mathlib are defined using filters [3] rather than the usual $\epsilon - \delta$ defini-

**Number Theory P2: ProverBench**

**Problem:** For the equation $x^2 + y^2 - 1 = 4xy$ its general solution in the integers is given by $x + u\sqrt{3} = (2 + \sqrt{3})^n$, where $u$ is the substitute for $y - 2x$.

```
import Mathlib

theorem general_solution_quadratic_equation (x y : ℤ) (u : ℤ
    ) (n : ℕ) :
  x^2 + y^2 - 1 = 4 * x * y → x + u * Real.sqrt 3 = (2 +
    Real.sqrt 3)^n :=
  sorry
```

*Figure 5.* Incorrect source statement: the correct Pell reduction gives $u + x\sqrt{3} = (2 + \sqrt{3})^n$.

tion, since filters unify the disparate conditions needed to define two-sided limits, one-sided limits, and limits to $\pm\infty$. Another example is that of a semilinear map (Dupuis et al., 2024), which is a generalization of linear maps, created to unify linear algebra over $\mathbb{R}$ and $\mathbb{C}$, where $\mathbb{C}$ has an extra complex-conjugate structure. These examples are not exhaustive; mathlib evolves daily, and it is a challenge to keep datasets up-to-date with emerging best practices and API.

**Fork proliferation.** *(Maintenance Decay)* Version drift cascades into fork proliferation, in which multiple ports of one dataset coexist with inconsistent splits, undocumented corrections, and unclear lineage. For miniF2F (Zheng et al., 2022) alone, widely used variants include the original release,[4] the Draft, Sketch, and Prove version (Jiang et al., 2023),[5] a Lean 4 port maintained by Kaiyu Yang,[6] the Kimina-Prover version (Wang et al., 2025),[7] and a version introduced by Harmonic (harmonic Community).[8] Papers often do not specify which version they used, making cross-paper comparisons unreliable.

**NL statement defect.** *(Benchmark Defects)* Some problems are defective before any formalization occurs. A ProverBench problem (Figure 5) states that the general solution to $x^2 + y^2 - 1 = 4xy$ is $x + u\sqrt{3} = (2 + \sqrt{3})^n$, whereas the correct Pell reduction gives $u + x\sqrt{3} = (2 + \sqrt{3})^n$, with the variables swapped. No amount of better Lean encoding can repair a wrong source problem; these defects require checking the mathematics itself.

---

[3] https://Leanprover-community.github.io/mathlib4_docs/Mathlib/Topology/Defs/Filter.

html
[4] https://github.com/openai/miniF2F
[5] https://github.com/facebookresearch/miniF2F
[6] https://github.com/yangky11/miniF2F-lean4/tree/main
[7] https://huggingface.co/datasets/AI-MO/miniF2F_test
[8] https://github.com/Harmonic-ai/datasets

# 4. Automated Checkers

We develop both static analyzers for mechanical hazards and LLM-based detectors for semantic mismatches. The two approaches are complementary: static checkers are cheap and deterministic but cannot assess whether a formalization captures mathematical intent; LLMs can reason about semantics but require human verification.

## 4.1. Static Checkers

We implement static analyzers as Lean 4 metaprograms (Paulino et al., 2024) that detect common formalization pitfalls. Our approach combines syntactic pattern matching with semantic guard proving: when a potentially problematic pattern is detected, we attempt to discharge the necessary guard condition using tactics like `omega`, `assumption`, and `simp`. A finding is reported only when the guard cannot be automatically proven. These serve as the equivalent to unit tests for Lean statements.

Table 3 summarizes our checkers, ordered by severity. The first three checkers target *soundness failures*: specifications that are provably wrong or vacuously true. The remaining checkers target issues arising from Lean's totalized arithmetic, where division by zero returns zero ($2/0 = 0$), imaginary components are converted to zero ($\sqrt{-1} = 0$), and natural subtraction truncates ($2 - 3 = 0$). Full descriptions appear in Appendix F.

*Table 3.* Automated checkers ordered by severity. Checkers marked with † support LLM-assisted false positive filtering.

| Checker | Description |
|---|---|
| Counterexample | Finds concrete values that disprove the theorem |
| Vacuous Theorem | Detects unsatisfiable hypotheses (trivially true) |
| Unsound Axiom | Use of `axiom` or `sorry` in proofs |
| Division by Zero† | Division, modulo, or inverse without non-zero guard |
| Nat Subtraction† | Natural subtraction that may truncate to zero |
| Analytic Domain† | Functions like `sqrt`, `log` outside valid domain |
| Unused Binder | Quantified variable not used in formula body |

**Corpus-scale audit results.** Running the static checkers over all released variants surfaces 4,833 findings and 399 proven issues (Table 4). These counts should not be read as deduplicated benchmark-level error rates: several rows are forks, ports, or subsets of the same underlying benchmark. They instead show that the audit is corpus-wide, while semantic equivalence still requires human adjudication.

**LLM-assisted false positive filtering.** Static analysis produces false positives when guards exist in forms the prover cannot recognize, for example, guards embedded in structure definitions, subtype constraints like $\{x : \mathbb{R} // 0 < x\}$, or non-local invariants. For findings in the †-marked categories, we employ a secondary LLM verification stage that examines the surrounding code context to determine whether a semantic guard exists.

The authors labeled a 55-example ProverBench warning-verification set spanning division by zero, Nat subtraction, analytic-domain, and modulo warnings (Table 5). Gemini 3.0 Flash achieves the highest accuracy (83.3%) at lowest cost. On ProverBench, LLM filtering reduces findings from 427 to 277 (35% reduction) while preserving all confirmed true positives. We conclude that although static checkers + LLM assisted filtering is not totally foolproof, they often successfully surface statements that are worth taking another look at.

## 4.2. LLM-Assisted Semantic Audit

While static checkers detect syntactic hazards, they cannot identify *semantic* mismatches between informal problems and their formal translations—missing hypotheses, incorrect domains, or definition mismatches. We evaluate whether LLMs can reliably detect such errors when prompted with our fault taxonomy.

**Detection framework.** For each benchmark problem, we present the LLM with the informal problem statement, the formal Lean statement, a description of one error category, and few-shot examples. The model outputs whether the specified error type is present. We evaluate six categories aligned with Table 2: problem statement errors, specification errors, formalization errors, domain mismatches, definition mismatches, and quantifier/indexing mismatches.

**Evaluation.** We evaluate on a curated 92-problem challenge set across FormalMath (22), ProofNet (13), ProverBench (23), and CombiBench (34), with each problem assessed across all six categories (552 total classifications). This set is not used to estimate benchmark-level prevalence. It is sourced from set of NL/Lean pairs for which we had ground-truth labels from community reports of various datasets like FormalMath and Proverbench, ProofNet/ProofNet# comparison, and Combibench diffs; unequal per-benchmark counts reflect where labeled errors were available.

Both models reach high recall but low precision overall (Table 6), and performance varies substantially by error type (Table 7): they detect **specification errors** and **definition mismatches**, where the discrepancy is often explicit, far more reliably than **formalization errors**, which require deeper semantic understanding of whether the Lean encoding preserves mathematical intent.

| Benchmark | Variant | Prob. | Find. | Proven | Div/0 | Nat Sub | Int Div | Analytic | Axiom | CEx |
|---|---|---|---|---|---|---|---|---|---|---|
| FormalMath | all | 5,560 | 3,250 | 141 | 1,127 | 582 | 423 | 351 | 0 | 55 |
| | lite | 425 | 213 | 6 | 80 | 34 | 25 | 33 | 0 | 1 |
| ProverBench | deepseek | 325 | 370 | 208 | 81 | 7 | 6 | 19 | 199 | 5 |
| ProofNet | original | 371 | 82 | 5 | 13 | 8 | 3 | 8 | 0 | 5 |
| | sharp (#) | 371 | 67 | 1 | 11 | 6 | 4 | 8 | 0 | 1 |
| miniF2F | v2c | 488 | 193 | 8 | 45 | 28 | 20 | 31 | 0 | 3 |
| | v2s | 488 | 181 | 8 | 45 | 29 | 15 | 28 | 0 | 3 |
| | yangky11 | 488 | 135 | 6 | 34 | 24 | 12 | 16 | 0 | 0 |
| | yangky11-early | 488 | 79 | 13 | 14 | 19 | 15 | 9 | 0 | 5 |
| | justincasher | 485 | 74 | 0 | 25 | 13 | 5 | 16 | 0 | 0 |
| | harmonic | 485 | 74 | 0 | 25 | 13 | 5 | 16 | 0 | 0 |
| | ai-mo | 244 | 37 | 1 | 17 | 6 | 3 | 5 | 0 | 0 |
| CombiBench | hf | 100 | 78 | 1 | 1 | 27 | 2 | 0 | 0 | 1 |
| **Total** | | **10,318** | **4,833** | **398** | 1,518 | 796 | 538 | 540 | 199 | 79 |

*Table 4.* Static-checker findings across all five benchmarks and their 13 released variants (∼10,000 problems; forks, ports, and splits are counted separately and not deduplicated). **Find.**: total audit signals. **Proven**: findings carrying a machine-checkable certificate of unprovability or vacuity. The category columns give the dominant hazard classes (**Div/0**, **Nat Sub**, **Int Div**, and **Analytic** are arithmetic/domain totalizations; **Axiom** and **CEx** are soundness failures); because a statement may trigger several checkers and a few smaller categories are omitted, the category columns need not sum to **Find.** or **Proven**. Upstream sources for every variant are given in Table 8 (Section A.2).

*Table 5.* LLM verification performance for filtering static checker false positives.

| Model | Accuracy | Precision | Recall | Cost |
|---|---|---|---|---|
| Gemini 3.0 Flash | 83.3% | 0.89 | 0.86 | $0.09 |
| GPT-5.2 | 81.8% | 0.92 | 0.83 | $0.18 |
| Claude Sonnet 4.5 | 81.5% | 0.89 | 0.81 | $0.42 |
| DeepSeek-V3 | 68.5% | 0.68 | 1.00 | $0.12 |

*Table 6.* LLM-assisted semantic audit: Models achieve high recall for detecting formalization errors but require human filtering of false positives.

| Model | Prec. | Rec. | F1 | Acc. | Cost |
|---|---|---|---|---|---|
| Sonnet 4.5 + Thinking | 0.30 | 0.82 | 0.42 | 68.1% | $13.71 |
| GPT-5.2 + Thinking | 0.24 | 0.91 | 0.37 | 54.6% | $6.86 |

*Table 7.* Per-type F1 scores for semantic error detection.

| Error Type | Sonnet 4.5 | GPT-5.2 |
|---|---|---|
| Problem Statement Error | 0.29 | 0.10 |
| Specification Error | 0.51 | 0.48 |
| Formalization Error | 0.18 | 0.15 |
| Domain Mismatch | 0.33 | 0.27 |
| Definition Mismatch | 0.57 | 0.54 |
| Quantifier/Indexing | 0.39 | 0.33 |

**Practical workflow.** We recommend a two-stage pipeline: (1) run static checkers to flag syntactic hazards cheaply and deterministically, then (2) apply LLM-based semantic review as a high-recall screen for problems that require semantic judgment. LLM-assisted audit can identify the ∼80% of problems likely to contain errors, allowing human experts to focus verification effort on flagged items. Human review remains the gold standard for final adjudication. Full per-dataset results and prompt details appear in Appendix H.

### 4.3. Effect on Reported Prover Scores

Defects matter for evaluation as they change reported numbers, and they can do so in two opposing ways. As a direct check, we selected 20 problems with mechanically proven issues across four datasets, manually corrected the statements, and evaluated provers on the original and corrected versions. The original flawed statements were unprovable, and both evaluated models solved 0/20; after correction, DeepSeek-Prover-V2-7B solved 3/20 and Kimina-Prover-8B solved 2/20. Such items do not make a benchmark easier; they silently *deflate* scores by adding impossible problems to the denominator.

The opposite failure mode also occurs: a formalization *weaker* than the informal problem is easier to prove and *inflates* scores. Consistent with this, repairing the weakened statements that distinguish ProofNet from its human-corrected counterpart ProofNet# (Poiroux et al., 2025) lowers measured pass rates for the provers we tested. Because the two effects pull in opposite directions, they can coexist within a single benchmark and partially cancel, leaving headline pass rates unreliable without a per-item dataset-quality audit.

# 5. Lessons Learned: Release Standards for Trustworthy Benchmarks

We propose release standards that are cheap to adopt and prevent common failure modes:

**Use `proof_wanted` for problem statements.** Lean's `sorry` tactic is widely used as a placeholder during benchmark creation, but it has a critical side effect: `sorry` adds the statement to the environment as an axiom, meaning any downstream code can reference it as a proven fact. An RL-trained prover can exploit this by citing the sorry-admitted statement rather than constructing a genuine proof. The `proof_wanted` keyword, introduced in Lean 4, solves this by declaring the theorem signature *without* adding it to the environment. The statement is visible for type-checking but cannot be used as a lemma, closing the sorry-exploitation loophole. Additionally, `proof_wanted` prevents the use of `native_decide` and other kernel-bypassing tactics, since no proof term is ever constructed. We recommend that all benchmark datasets use `proof_wanted` instead of `sorry` for unsolved problem statements.

**Turn off auto-implicit.** Most benchmarks are created by auto-formalizing natural language with LLMs. When the LLM mistypes a variable name or forgets to define a type parameter, Lean's auto-implicit feature silently "fixes" it by adding implicit parameters. Lean's default autoImplicit automatically inserts hidden type parameters and universe levels for undeclared symbols, which is ergonomic for humans but risky for NL→Lean pipelines. The file type-checks, so nobody notices. The error only surfaces later when one tries to prove it and finds it either trivial or impossible. Adding `set_option autoImplicit false` makes these errors fail at formalization time instead of hiding them. This forces errors at formalization time until all intended domains and binders are declared explicitly.

**Basic Checkers: Be disciplined about types and common arithmetic traps.** A minimal checker pass should flag known Lean pitfalls[9].

**Avoid using Axioms.** In Lean, writing `axiom h :` $\phi$ (or `constant h :` $\phi$) asserts a proof of $\phi$ exists; any solver can then "solve" the item by citing h, which collapses evaluation and obscures whether the statement is even satisfiable. We should therefore adopt a no-axioms rule for dataset files: problem statements are encoded as definitions of propositions, never as axioms.

**Dataset Maintenance and Pinning.** When datasets are released, the Lean/mathlib version they are intended for should also be made clear, so it serves as a reference when being solved by a model in more recent Lean versions. Re-

gardless of if the dataset is autoformalized or human written, there isn't always a mathematical equivalent in Lean and it is an evolving language so some definitions and ease of translation will change over time.

**Capture all requirements, not just the main claim.** While human verification is the gold standard for identifying errors in formalization, it is costly and time-consuming. LLM-as-a-judge can be used as a triage tool to identify whether a formalization captures all requirements in a given statement rather than only the main claim.

# 6. Related Work

**Formal math benchmarks and repair efforts.** Standard formal mathematics benchmarks include *miniF2F* (Zheng et al., 2022), *ProofNet* (Azerbayev et al., 2023), *FormalMath* (Yu et al., 2025), *CombiBench* (Liu et al., 2025a), and *ProverBench* (Ren et al., 2025). The closest repair-oriented work is miniF2F-v2 (Ospanov et al., 2025), which performs a detailed human audit of miniF2F, documents discrepancies between formal and informal statements, and releases repaired statements and proofs. Our work is complementary: rather than presenting a fully repaired single benchmark, we audit a broader benchmark suite and focus on a taxonomy, scalable static checks, and release standards that help direct expert effort.

Recent work on the Erdős problems database has highlighted both the promise and perils of AI-assisted formalization at scale (Alexeev, 2025). Alexeev developed an automated pipeline combining ChatGPT for proof explanation with Harmonic's Aristotle system (Achim et al., 2025) for autoformalization, successfully formalizing solutions to approximately ten problems with minimal human intervention. Notably, Aristotle autonomously resolved Problem 124 by exploiting a weaker formulation than Erdős intended, a misformalization that went undetected until the AI found a trivial proof. This experience underscores our findings: the same AI systems that accelerate formalization can also serve as effective detectors of specification errors, discovering missing hypotheses (Problem 56), incorrect variable constraints (Problem 480 where $m \neq 0$ was written instead of $n \neq 0$), and vacuous hypotheses.

**Autoformalization and semantic-fidelity evaluation.** The task of translating informal mathematical statements into formal specifications has been studied extensively (Wu et al., 2022; Jiang et al., 2023). Recent work demonstrates that LLMs can produce syntactically correct Lean code at scale, though semantic fidelity remains challenging (Poiroux et al., 2025; Liu et al., 2025b; Lu et al., 2025). Liu et al. (2025c) scale theorem-statement generation through lifting, augmentation, and synthesis; Liu et al. (2025b) propose a formal-grounded equivalence metric and dependency re-

---

[9] https://leanprover-community.github.io/extras/pitfalls.html

trieval for statement autoformalization; Lu et al. (2025) evaluate alignment between informal and formal statements; and Cabral et al. (2026) focus on preserving proof-step structure via dependency graphs. These works improve generation or evaluation of formalizations. Our focus is orthogonal: we study benchmark defects and evaluation loopholes after statements enter the theorem-proving evaluation pipeline.

**Specification errors in formal verification.** The software verification community has long recognized that formal proofs guarantee correctness only relative to their specifications, and specification errors can be as dangerous as implementation bugs (Woodcock et al., 2009). Studies of industrial verification efforts report that specification defects account for a substantial fraction of issues discovered during proof attempts (Klein et al., 2014). The seL4 microkernel verification, for instance, required extensive iteration to align formal specifications with intended behavior (Klein et al., 2009). Our work applies these lessons to the ML benchmarking setting, where the pressure to scale dataset creation amplifies specification risk.

**Counterexample generation.** Automated counterexample finding has proven effective for detecting specification errors in proof assistants. Isabelle's Nitpick tool encodes higher-order logic into first-order relational logic and invokes SAT solvers to find finite countermodels (Blanchette & Nipkow, 2010), while Quickcheck provides randomized testing (Bulwahn, 2012). Adapting these techniques to dependent type theory remains challenging due to the richer type structure; recent progress includes Chako, which integrates Lean with the Nunchaku model finder (Reynolds et al., 2016). Our vacuity and counterexample checkers draw on this tradition, though many benchmark defects still require semantic understanding beyond what current model finders can provide.

## 7. Limitations

We do not claim a complete human-verified enumeration of all defects across all items in all datasets and their forks. Some issues require mathematical insight (e.g., whether a question is false or vacuous) or Lean expertise (e.g., translating mathematics into Lean with the correct encoding). Our goal is to (i) demonstrate that defects are common enough to matter, (ii) propose checks and standards that prevent high-impact failures, and (iii) make future benchmark development easier by releasing checkers and prompts.

Our implementation is Lean-specific, but the taxonomy is not. Fidelity failures arise whenever natural-language mathematics is translated into a formal language; evaluation loopholes arise whenever a solver is rewarded for passing an automated checker; and maintenance decay arises when-

ever libraries, proof assistants, or benchmark forks evolve. Other proof assistants would require different static checks, for example different arithmetic totalization rules or axiom policies, but the audit structure transfers.

## 8. Conclusion & Future Work

Lean proof checking removes many sources of error in theorem proving evaluation, but it does not guarantee benchmark validity. Specification fidelity is a known challenge across formal methods and ML evaluation, yet systematic audits of Lean theorem-proving benchmarks have been lacking: a gap our taxonomy, checkers, and case studies aim to fill.

Future work involves extending this to more benchmarks, generating better high-quality synthetic data and filtering existing lean datasets, and developing an effective theorem proving harness.

## Software and Data

Our static checkers (implemented as Lean 4 metaprograms), the evaluation harness, the semantic-audit prompts, and snapshots of the audited benchmark variants are available at `https://github.com/Shashi456/atp-checkers`.

## Acknowledgements

We gratefully acknowledge support from Renaissance Philanthropy's AI for Math Fund through the MathBench: Towards Evaluating Natural Language Proofs project. We thank the AI for Math Fund team for supporting open research infrastructure at the intersection of artificial intelligence and mathematics. The views expressed in this paper are those of the authors and do not necessarily reflect the views of the funders.

## Impact Statement

This paper presents work aimed at advancing the field of formal mathematics benchmarking. By identifying systematic issues in current benchmarks, we aim to improve the reliability of evaluations in automated theorem proving, leading to a more accurate assessment of progress in the field. Our checkers and taxonomy can help benchmark creators catch defects before release, reducing the risk that published results reflect evaluation artifacts rather than genuine capability. The release of our static checkers and LLM prompts may also improve the quality of training data for formal reasoning systems. More broadly, documenting failure modes in formal verification pipelines enhances the trustworthiness of formally verified software, with implications for safety-critical systems. We also release these tools, hop-

ing that they will be helpful in improving the data quality of pre-training and post-training. As automated theorem provers are increasingly used to verify mathematical claims and software correctness, the integrity of their evaluation infrastructure becomes a matter of broader scientific trust. We hope this work contributes to a culture of rigorous benchmark stewardship in the formal methods community.

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

# A. Dataset Details

Our goal is to demonstrate that defects are systematic, not anecdotal, and to identify checks that catch them early.

## A.1. Popular Formal Math Datasets

We focus on widely used Lean datasets that collectively represent the standard evaluation suite for neural theorem provers:

1. **miniF2F** (Zheng et al., 2022): 488 Olympiad-level problems (244 validation, 244 test), originally released in Lean 3.

2. **ProofNet** (Azerbayev et al., 2023): 371 undergraduate-level problems from textbooks (analysis, algebra, topology), released in Lean 3 with paired natural-language statements and proofs.

3. **FormalMath** (Yu et al., 2025): 5,560 Lean 4 problems spanning from Olympiad challenges to undergraduate-level theorems across algebra, applied mathematics, calculus, number theory, and discrete mathematics. They also release FormalMath Lite, a carefully selected subset of 425 problems (comprising 359 high school-level and 66 undergraduate-level problems).

4. **CombiBench** (Liu et al., 2025a): A benchmark comprising 100 combinatorial problems, each formalized in Lean 4 and paired with its corresponding informal statement.

5. **ProverBench** (Ren et al., 2025): A benchmark dataset comprising 325 problems. 15 are formalized from AIME 24 and 25. The remaining 310 problems are drawn from curated textbook examples and educational tutorials.

## A.2. Provenance of Audited Variants

Because ports, forks, and corrected re-releases of these benchmarks proliferate (Section 3), and because papers rarely record which one they evaluated, we list the exact upstream source of every variant audited in Table 4. The slugs in Table 8 match the **Variant** column of that table. A few details warrant a note. The miniF2F v2c and v2s entries are the "complete" and "statement" splits of the re-verified miniF2F-v2 release (Ospanov et al., 2025). The justincasher and harmonic rows are the *same* corrected formalizations (Harmonic's re-release) packaged under different train/validation/test partitions; because the static checkers range over the entire file, their counts in Table 4 coincide, and we list both because each is distributed and evaluated as a separate artifact. The yangky11 and yangky11-early rows are two commits of Kaiyu Yang's miniF2F-lean4 repository: the current main (commit 6306b5f) and an earlier commit (7a2d40b, dated 2023-11-17) that predates its later statement fixes.

# B. Issues in Formal Mathematics Datasets

This appendix has examples of various formalization issues found across major formal mathematics benchmarks. We present side-by-side comparisons of problematic formalizations, and describe the error and either point out which version is correct or provide the correction. This section is meant to familiarize the reader with the kinds of pitfalls we observe in datasets. This is not a complete list of all errors in datasets like miniF2F, ProofNet, FormalMath, ProverBench which would span 100s of problems if not 1000s.

## B.1. Issue 1: Incomplete Formalization

Many problems in formal benchmarks only formalize part of the original problem, often missing crucial constraints or conditions.

### B.1.1. IMO 1983 PROBLEM 6 - MINIF2F

In this problem, the solution must first prove the inequality and then determine when equality occurs.

> **Problem Statement**
>
> **Problem:** Let $a$, $b$ and $c$ be the lengths of the sides of a triangle. Prove that $a^2b(a-b) + b^2c(b-c) + c^2a(c-a) \geq 0$. Determine when equality occurs.

*Table 8.* Provenance of the 13 audited variants in Table 4. `hf` denotes a Hugging Face release; where a variant mirrors a canonical repository, both are given.

| Benchmark | Variant | Upstream source |
|-----------|---------|-----------------|
| FormalMATH | `all` | https://huggingface.co/datasets/SphereLab/FormalMATH-All |
| FormalMATH | `lite` | https://huggingface.co/datasets/SphereLab/FormalMATH-Lite project: https://github.com/Sphere-AI-Lab/FormalMATH-Bench |
| ProverBench | `deepseek` | https://huggingface.co/datasets/deepseek-ai/DeepSeek-ProverBench |
| ProofNet | `deepseek` | https://github.com/deepseek-ai/DeepSeek-Prover-V1.5/blob/main/datasets/proofnet.jsonl |
| ProofNet | `sharp` (#) | https://huggingface.co/datasets/PAug/ProofNetSharp |
| miniF2F | `v2c`, `v2s` | https://github.com/roozbeh-yz/miniF2F_v2 (complete / statement splits) |
| miniF2F | `yangky11` | https://github.com/yangky11/miniF2F-lean4, commit `6306b5f` |
| miniF2F | `yangky11-early` | https://github.com/yangky11/miniF2F-lean4, commit `7a2d40b` (2023-11-17) |
| miniF2F | `justincasher` | https://github.com/justincasher/miniF2F (Harmonic's formalizations, original miniF2F splits) |
| miniF2F | `harmonic` | https://github.com/harmonic-ai/datasets/tree/main/minif2f (same formalizations, Harmonic's own splits) |
| miniF2F | `ai-mo` | https://huggingface.co/datasets/AI-MO/minif2f_test |
| CombiBench | `hf` | https://huggingface.co/datasets/AI-MO/CombiBench canonical: https://github.com/MoonshotAI/CombiBench |

**DSP/OpenAI/Numina Version**

```
1  theorem imo_1983_p6
2    (a b c : ℝ)
3    (h₀ : 0 < a ∧ 0 < b ∧ 0 < c)
4    (h₁ : c < a + b)
5    (h₂ : b < a + c)
6    (h₃ : a < b + c) :
7    0 ≤ a^2 * b * (a − b) +
8        b^2 * c * (b − c) +
9        c^2 * a * (c − a) :=
10 begin
11   sorry
12 end
```

**Harmonic Version (Correct)**

```
1  theorem formal_1647
2    (a b c d : ℝ)
3    (h₀ : 0 < a ∧ 0 < b ∧ 0 < c)
4    (h₁ : c < a + b)
5    (h₂ : b < a + c)
6    (h₃ : a < b + c)
7    (h₄ : d = a^2 * b * (a − b) +
8          b^2 * c * (b − c) +
9          c^2 * a * (c − a)) :
10   0 ≤ d ∧ (d = 0 ↔ (a = b ∧ b = c))
     := by
11   sorry
```

B.1.2. IMO 1981 PROBLEM 6 - MINIF2F

The problem asks to "Determine $f(4, 1981)$" – a specific computational problem requiring finding an actual value. While the formalization asks for a recurrence relation $f(4, y + 1) = 2^{f(4,y)+3} - 3$. While it might be the case that the recurrence relation might be used to solve the computational problem, this is not an exact formalization. Proving the recurrence is necessary but not sufficient for computing the numerical value.

**Problem Statement**

**Problem:** The function $f(x, y)$ satisfies: (1) $f(0, y) = y + 1$, (2) $f(x + 1, 0) = f(x, 1)$, (3) $f(x + 1, y + 1) = f(x, f(x + 1, y))$, for all non-negative integers $x, y$. Determine $f(4, 1981)$.

**DSP version**

```
1  theorem imo_1981_p6
2    (f : ℕ → ℕ → ℕ)
3    (h₀ : ∀ y, f 0 y = y + 1)
4    (h₁ : ∀ x, f (x + 1) 0 = f x 1)
5    (h₂ : ∀ x y, f (x + 1) (y + 1) =
6          f x (f (x + 1) y)) :
7    ∀ y, f 4 (y + 1) =
8    2^(f 4 y + 3) − 3 :=
9  begin
10   sorry
11 end
```

**Harmonic version**

```
1  theorem formal_1514
2    (f : ℕ → ℕ → ℕ)
3    (h₀ : ∀ y, f 0 y = y + 1)
4    (h₁ : ∀ x, f (x + 1) 0 = f x 1)
5    (h₂ : ∀ x y, f (x + 1) (y + 1) =
6          f x (f (x + 1) y)) :
7    ∀ y, f 4 (y + 1) =
8    2^(f 4 y + 3) − 3 := by
9    sorry
```

### B.1.3. AXLER EXERCISE 3.8 - PROOFNET

The formalization is missing the properties of V where it should be finite dimensional. The existence of complements/kernel intersections depends on finite-dimensionality; otherwise $U \cap \ker T = \{0\}$ with range $T = T(U)$ need not hold.

**Problem Statement**

**Problem:** Suppose that $V$ is finite dimensional and that $T \in \mathcal{L}(V, W)$. Prove that there exists a subspace $U$ of $V$ such that $U \cap \text{null } T = \{0\}$ and range $T = \{Tu : u \in U\}$.

**ProofNet - Axler Exercise 3.8 (Missing finite dimension hypothesis)**

```
1  theorem exercise_3_8 {F V W : Type*} [add_comm_group V]
2    [add_comm_group W] [field F] [module F V] [module F W]
3    (L : V →[F] W) :
4    ∃ U : submodule F V, U ⊓ L.ker = ⊥ ∧
5    linear_map.range L = range (dom_restrict L U):=
```

## B.2. Issue: Missing Specification

### B.2.1. IMO 1962 PROBLEM 2 - MINIF2F

The DSP/numina version doesn't specify that $\sqrt{3-x} - \sqrt{x+1} > 0$, and while it might be perhaps immaterial while solving the problem, it is a case of missing specification and leaving nothing to ambiguity.

**Problem Statement**

**Problem:** Show that if the real number $x$ satisfies the inequality $\sqrt{\sqrt{3-x} - \sqrt{x+1}} < \frac{1}{2}$, then $-1 \le x < 1 - \frac{\sqrt{127}}{32}$.

**DSP/Numina version**

```
1  theorem imo_1962_p2
2    (x : ℝ)
3    (hₓ : 0 ≤ 3 - x)
4    (h₁ : 0 ≤ x + 1)
5    (h₂ : 1 / 2 < real.sqrt (3 - x) -
6        real.sqrt (x + 1)) :
7    -1 ≤ x ∧ x < 1 -
8    real.sqrt 31 / 8 :=
9  begin
10   sorry
11 end
```

**Harmonic version**

```
1  theorem formal_4518
2    (x : ℝ)
3    (hₓ : x ≤ 3)
4    (h₁ : -1 ≤ x)
5    (h₂ : 0 ≤ Real.sqrt (3 - x) -
6        Real.sqrt (x + 1))
7    (h₃ : Real.sqrt (Real.sqrt (3 - x) -
8        Real.sqrt (x + 1)) > 1 / 2) :
9    -1 ≤ x ∧ x < 1 -
10   Real.sqrt 127 / 32 := by
11   sorry
```

### B.2.2. APMO 1991 PROBLEM 2 - COMBIBENCH

In this problem originally, the points were not assumed to be distinct, so it was trivially false. It was later fixed.

**Problem Statement**

**Problem:** Suppose there are 997 points given in a plane. If every two points are joined by a line segment with its midpoint coloured in red, show that there are at least 1991 red points in the plane.

**Version 1**

```
1  import Mathlib
2
3  noncomputable def red_points {k}
4    (points : Fin k → ℝ × ℝ) :
5    Finset (ℝ × ℝ) :=
6    ((Finset.univ (α := Fin k × Fin k)).
       image
7      (fun x => midpoint ℝ (points x.1)
                            (points x.2)))
8
9
10 theorem apmo_1991_p2
11   (points : Fin 997 → ℝ × ℝ) :
12   (red_points points).card ≥ 1991 :=
     by
13   sorry
```

**Fixed**

```
1  import Mathlib
2
3  noncomputable def red_points {k}
4    (points : Fin k → ℝ × ℝ) :
5    Finset (ℝ × ℝ) :=
6  (((Finset.univ (α := Fin k × Fin k)) \
7    (Finset.univ).image (fun i => (i, i))
       ).image
8      (fun x => midpoint ℝ (points x.1)
9                           (points x.2)))
10
11 theorem apmo_1991_p2 (points : Fin 997
         → ℝ × ℝ)
12   (hpoints : Function.Injective points)
       :
13   (red_points points).card ≥ 1991 :=
     by
14   sorry
```

## B.3. Issue: Incorrect Translation

Some formalizations fundamentally misrepresent the original problem, leading to different mathematical statements.

### B.3.1. MATHD ALGEBRA 15 - MINIF2F

In this case, Harmonic version doesn't capture the general operation definition from the problem statement instead it directly introduces the substituted weaker version of the problem to solve. DSP and OpenAI versions define a general operation $s$ while Harmonic directly computes the specific case.

**Problem Statement**

**Problem:** If $a * b = a^b + b^a$, for all positive integer values of $a$ and $b$, show that $2 * 6 = 100$

**DSP Version**

```
1  theorem mathd_algebra_15
2    (s : ℕ → ℕ → ℕ)
3    (h₀ : ∀ a b, 0 < a ∧ 0 < b →
4     s a b = a^(b:ℕ) + b^(a:ℕ)) :
5    s 2 6 = 100 :=
6  begin
7    rw h₀,
8    refl,
9    norm_num,
10 end
```

**Harmonic Version**

```
1  theorem formal_2895 :
2    2 ^ 6 + 6 ^ 2 = 100 := by
3    sorry
```

**OpenAI Version**

```
1  theorem mathd_algebra_15
2    (s : ℕ → ℕ → ℕ)
3    (h₀ : ∀ a b, 0 < a ∧ 0 < b →
4     s a b = a^(b:ℕ) + b^(a:ℕ)) :
5    s 2 6 = 100 :=
6  begin
7    rw h₀,
8    refl,
9    norm_num,
10 end
```

### B.3.2. NUMBER THEORY PROBLEM 2 - PROVERBENCH

In this, problem u is a free variable and the specification that u is a substitute for $y - 2x$ is missing. Not only that, upon further look there might be an error in the problem, where $x + u\sqrt{3}$ does not solve the question, and the actual problem should have $u + x\sqrt{3}$.

**Problem Statement**

**Problem:** For the equation $x^2 + y^2 - 1 = 4xy$ its general solution in the integers is given by $x + u\sqrt{3} = (2 + \sqrt{3})^n$, where $u$ is the substitute for $y - 2x$.

**ProverBench Formalization (Incorrect)**

```
1  import Mathlib
2
3  theorem general_solution_quadratic_equation (x y : ℤ) (u : ℤ) (n : ℕ) :
4    x^2 + y^2 - 1 = 4 * x * y → x + u * Real.sqrt 3 = (2 + Real.sqrt 3)^n :=
5    sorry
```

### B.4. Issue: Wrong Specification

### B.4.1. QUANTITATIVE_REASONING_ZH_BLUE_41 - FORMALMATH PROBLEM 5164

In this problem, truncated subtraction occurs due to natural numbers, and this is incorrect typecasting.

**Problem Statement**

**Problem:** Given $M = \{x \mid x = a^2 + 1, a \in \mathbb{N}^*\}$, $N = \{x \mid x = b^2 - 4b + 5, b \in \mathbb{N}^*\}$, then the relationship between $M$ and $N$ is Proof: The answer is $M \subseteq N$. From $a^2 + 1 = (a + 2)^2 - 4a + 5$, we can see that $M \subseteq N$. However, $1 \in N$ and $1 \notin M$.

---

**FormalMath 5164**

```
1  import Mathlib
2
3  open Set Real
4  open scoped BigOperators
5
6  theorem quantitative_reasoning_zh_blue_41 :
7    {x : ℕ | ∃ a : ℕ, 0 < a ∧ x = a^2 + 1} ⊂ {x : ℕ |
8    ∃ b : ℕ, 0 < b ∧ x = b^2 − 4 * b + 5} := by
9    sorry
```

---

### B.4.2. PROBLEM 48 - FORMALMATH

This does not avoid the case that n can take the value 0.

---

**Problem Statement**

**Problem:** If the sequence $\{a_n\}$ satisfies that for any $n \in \mathbb{N}^*$, $\sum_{d|n} a_d = 2^n$, prove that $n \mid a_n$.

---

**FormalMath Formalization**

```
1  theorem algebra_56552 (a : ℕ → ℕ) (ha : ∀ n, ∑ d in n.divisors, a d = 2 ^ n) :
2    ∀ n, n | a n := by
3    sorry
```

---

## C. Fault Taxonomy: Detailed Definitions

This appendix provides detailed definitions and examples for each error category in our taxonomy.

### C.1. Specification Errors vs. Formalization Errors

The distinction between specification and formalization errors is crucial for understanding benchmark defects.

**Specification errors** occur when the formalization is *incomplete*—it omits content that should be present. The Lean statement is a proper subset of what the informal problem requires:

- **Missing hypotheses**: The informal problem states conditions not encoded in Lean. Example: "Suppose $V$ is finite-dimensional" becomes a theorem about general vector spaces.

- **Missing subgoals**: Multi-part problems have some parts dropped. Example: "Prove $P$ and determine when equality holds" becomes a theorem proving only $P$.

- **Underspecified constraints**: Implicit mathematical conventions are not made explicit. Example: "for positive integers" uses `(n : ℕ)` without `(hn : 0 < n)`.

A specification error in isolation typically makes a problem *easier* (fewer constraints, fewer subgoals to prove).

**Formalization errors** occur when the translation *misrepresents* the original—not by omission, but by encoding something different:

- **Incorrect translation**: The mathematical content is mistranslated. Example: "$x$ divides $y$" encoded as $x/y$ (division) instead of $x \mid y$ (divisibility).

- **Encoding artifacts**: The translation introduces structure not in the original. Example: encoding an existential with a specific witness constrains solutions.

- **Logic errors**: Wrong quantifier scope, implication direction, or connectives. Example: $\forall x, P(x) \rightarrow \exists y, Q(y)$ vs. $\exists y, \forall x, P(x) \rightarrow Q(y)$.

A formalization error may make a problem easier, harder, or *impossible* to prove correctly.

**Remediation differs.**

- Specification errors require *adding* content: insert missing hypotheses, add subgoals, make constraints explicit.

- Formalization errors require *correcting* content: fix mistranslations, remove artifacts, restructure logic.

### C.2. Domain and Type Mismatches

A special class of formalization errors where the mathematical domain is incorrectly encoded:

**Wrong number system.** Using $\mathbb{N}$ when $\mathbb{Z}$ or $\mathbb{R}$ is intended. This is particularly dangerous because:

- Nat subtraction truncates: `5 - 7 = 0`

- No negative numbers: statements about "all integers" become statements about naturals

- Division semantics differ across types

**Incorrect type structure.**

- Using `List` when `Set` or `Finset` is needed (ordering, duplicates)

- Encoding bijections as two functions instead of `Equiv`

- Using `Type` when `Prop` would be more appropriate

**Coercion hazards.**

- `Int`.toNat truncates negatives to 0

- Coercing $\mathbb{N}$ to $\mathbb{R}$ loses decidability of equality

- Implicit coercions can hide type mismatches

### C.3. Definition Mismatches

The formalization uses the wrong mathematical definition:

**Outdated definitions.** Using hand-rolled definitions when mathlib provides standard ones. Example: defining "perfect set" as "closed and every point is a cluster point" instead of using `Perfect`.

**Subtly different concepts.**

- `IsClosed` vs. `IsCompact` vs. `Perfect`

- `Continuous` vs. `Differentiable` vs. `ContDiff`

- `Subgroup` vs. `Submonoid` vs. `Subsemigroup`

**Wrong level of generality.**   Stating a theorem about groups when it requires rings, or over-specializing to $\mathbb{R}$ when the result holds for any ordered field.

## C.4. Quantifier and Indexing Mismatches

**Quantifier scope errors.**   Whether a variable is bound at statement level or inside the formula:

- $\forall \epsilon > 0, \exists \delta$ (correct for limits)

- $\exists \delta, \forall \epsilon > 0$ (wrong—$\delta$ cannot depend on $\epsilon$)

**Indexing conventions.**   Mathematical statements often use 1-indexed sequences (“$a_1, a_2, \ldots, a_n$”); Lean's `Fin n` and `List` are 0-indexed. Off-by-one errors can make statements false or vacuously true.

**Unused binders.**   Quantified variables that don't appear in the formula body:

```
theorem problem (n : ℕ) : IsLeast {k | ...} 100   -- n unused!
```

Often indicates copy-paste errors or incomplete translation.

## C.5. Lean Encoding Hazards

These are syntactically identifiable patterns that can silently change semantics:

**Natural number subtraction.**   In $\mathbb{N}$, subtraction is truncated: `a - b = 0` when $b \geq a$. This affects:

- Difference expressions: `n - 1` is 0 when `n = 0`

- Loop bounds: iterating from `n - k` to `n` may iterate 0 times

- Inequalities: `a - b < c` behaves unexpectedly near zero

**Division and modulo by zero.**   In mathlib fields and division rings:

- `a / 0 = 0` (not undefined)

- $0^{-1} = 0$

- `n % 0 = n` for natural numbers

Unguarded denominators can trivialize goals or make false statements provable.

**Totalized analytic functions.**   Mathlib totalizes partial functions with junk values:

- `Real.sqrt x = 0` for $x < 0$

- `Real.log x = 0` for $x \leq 0$

- `Real.arcsin x = 0` for $|x| > 1$

## C.6. Evaluation Loopholes

**Axiom injection.**   If the environment contains `axiom h : P`, any solver can prove $P$ by citing `h`. This collapses evaluation—we cannot tell if the model found a real proof.

**The `apply?` bug.**   In Lean $<$4.20.0, `apply?` could introduce a synthetic sorry without logging errors. Combined with universe-level mismatches, this caused Lean to report success while the theorem was never kernel-verified. See Appendix D for examples.

**Overly powerful automation.**

- `native_decide` can brute-force decidable goals

- **`decide`** on incorrectly-typed goals can succeed trivially

- Unrestricted **`simp`** with `[*]` may exploit unintended lemmas

### C.7. Maintenance and Source Issues

**Source validity problems.**

- Defective NL statements (typos, ambiguity, mathematical errors)

- False claims (the theorem is actually not true)

- Ambiguous specifications admitting multiple interpretations

**Version drift.**

- Lean 3 $\rightarrow$ 4 migration changes syntax and semantics

- Mathlib API changes rename or restructure definitions

- Tactic behavior changes between Lean versions

**Fork proliferation.** Multiple versions of datasets coexist with:

- Inconsistent train/val/test splits

- Undocumented corrections (some forks fix bugs, others don't)

- Unclear lineage (which version is upstream?)

Papers rarely specify which fork they used, making cross-paper comparisons unreliable.

## D. The `apply?` Frontend Bug: Example Exploits

The following proofs, generated by DeepSeek-Prover-V2-7B on PutnamBench, trigger a frontend bug in Lean 4.9.0 where `apply?` combined with a universe-level error causes Lean to silently drop the theorem without kernel verification. The bug occurs because `apply?` calls `admitGoal` without logging errors, creating a synthetic sorry. When elaboration encounters a universe parameter error, it stops before `addDecl` is reached, so the expected "declaration uses sorry" warning is never emitted.

---

**Problem Statement**

**Putnam 2005-A4:** Let $H$ be an $n \times n$ matrix all of whose entries are $\pm 1$ and whose rows are mutually orthogonal. Suppose $H$ has an $a \times b$ submatrix whose entries are all $1$. Show that $ab \leq n$.

---

**DeepSeek-Prover-V2-7B Output (Buggy)**

```
 1  theorem putnam_2005_a4
 2      (n : ℕ) (H : Matrix (Fin n) (Fin n) ℝ) (a b : ℕ)
 3      (S : Matrix (Fin a) (Fin b) ℝ) (npos : n ≥ 1)
 4      (Hentries : ∀ i j : Fin n, H i j = 1 ∨ H i j = -1)
 5      (Hortho : H.HasOrthogonalRows)
 6      (hab : 1 ≤ a ∧ a ≤ n ∧ 1 ≤ b ∧ b ≤ n)
 7      (Ssub : ∃ (rri : Fin a → Fin n) (cri : Fin b → Fin n),
 8              rri.Injective ∧ cri.Injective ∧ S = H.submatrix rri cri)
 9      (Sentries : ∀ (i : Fin a) (j : Fin b), S i j = 1)
10      : a * b ≤ n := by
11    obtain ⟨rri, cri, hrinj, hcrij, hsub⟩ := Ssub
12    have h₁ : a ≤ n := hab.2.1
13    have h₂ : b ≤ n := hab.2.2.2
14    have h₃ : a * b ≤ n := by
15      calc
16        a * b = Cardinal.toNat (a * b) := by simp [Nat.cast_inj]
17        _ ≤ Cardinal.toNat (n) := by
18        apply?  -- BUG: introduces synthetic sorry without error
19        _ = n := by simp [Nat.cast_inj]
20    exact h₃
```

**Problem Statement**

**Putnam 2007-B4:** Let $n$ be a positive integer. Find the number of pairs $P, Q$ of polynomials with real coefficients such that $(P(X))^2 + (Q(X))^2 = X^{2n} + 1$ and $\deg P > \deg Q$.

**DeepSeek-Prover-V2-7B Output (Buggy)**

```
 1  theorem putnam_2007_b4 (n : ℕ) (npos : n > 0) :
 2      ({(P, Q) : (Polynomial ℝ) × (Polynomial ℝ) |
 3        P ^ 2 + Q ^ 2 = Polynomial.X ^ (2 * n) + 1 ∧
 4        P.degree > Q.degree}.ncard = ((fun n ↦ 2 ^ (n + 1)) : ℕ → ℕ) n) := by
 5    have h₁ : ∀ n : ℕ, n > 0 →
 6        ({(P, Q) : (Polynomial ℝ) × (Polynomial ℝ) |
 7          P ^ 2 + Q ^ 2 = Polynomial.X ^ (2 * n) + 1 ∧
 8          P.degree > Q.degree}.ncard : ℕ) = 2 ^ (n + 1) := by
 9      intro n hn
10      have h : ∀ n : ℕ, n > 0 → ... := by
11        intro n hn
12        have h₁ : ... := by
13          have h₂ : ... := by
14            rw [← Cardinal.natCast_inj]
15            apply?  -- BUG: introduces synthetic sorry without error
16          exact_mod_cast h₂
17        exact h₁
18      exact h n hn
19    have h₂ : ... := by apply h₁; exact npos
20    simpa [h₂] using h₂
```

In both cases, the `apply?` tactic introduces a synthetic sorry that, combined with the `Cardinal` universe coercion, triggers the bug. Lean 4.9.0 reports no errors, but the theorems are never actually verified by the kernel. This bug was fixed in Lean 4.20.0.[10]

---

[10] https://github.com/leanprover/lean4/pull/8231

## E. Vacuous Hypotheses: An Example from CombiBench

The following example, taken from an earlier version of CombiBench (Liu et al., 2025a) (since corrected), illustrates how incorrect translation can produce vacuous hypotheses that make a statement trivially provable.

---

**Problem Statement**

**IMOSL 2011-C6:** Let $n$ be a positive integer and let $W$ be an infinite periodic word consisting of letters $a$ and $b$, with minimal period $N > 2^n$. A finite nonempty word $U$ is *ubiquitous* if all four words $Ua$, $Ub$, $aU$, and $bU$ appear in $W$. Prove that there are at least $n$ ubiquitous finite nonempty words.

---

**CombiBench Formalization (Vacuous Hypotheses)**

```
1  def appears (W : ℤ → Fin 2) (U : Σ n, Fin n → Fin 2) : Prop :=
2    ∃ k, ∀ i : Fin U.1, U.2 i = W (k + i)
3
4  def ubiquitous (W : ℤ → Fin 2) (U : Σ n, Fin n → Fin 2) : Prop :=
5    appears W ⟨U.1 + 1, Fin.snoc U.2 0⟩ ∧
6    appears W ⟨U.1 + 1, Fin.snoc U.2 1⟩ ∧
7    appears W ⟨U.1 + 1, Fin.cons 0 U.2⟩ ∧
8    appears W ⟨U.1 + 1, Fin.cons 1 U.2⟩
9
10 theorem imosl_2011_c6 (W : ℤ → Fin 2) (n : ℕ+) (N : ℕ) (hN : 2 ^ n.1 < N)
11     (hW : Function.Periodic W N)
12     (hW' : ∀ N' < N, ¬Function.Periodic W N') :   -- problematic!
13     ∃ (x : Fin n ↪ (Σ k, Fin k → Fin 2)),
14       (∀ i, (x i).1 ≠ 0) ∧ (∀ i, ubiquitous W (x i)) := by
15     sorry
```

---

**The problem.** The hypothesis hW' attempts to encode that $N$ is the *minimal* period: no smaller value $N' < N$ should also be a period. However, Function.Periodic W 0 is *definitionally true* in Lean—it unfolds to $\forall z, W(z+0) = W(z)$, which reduces to $\forall z, W(z) = W(z)$ and is provable by rfl.

Since $n \geq 1$ implies $2^n \geq 2$, and $N > 2^n$, we have $N \geq 3$, so $0 < N$. Therefore hW' 0 yields ¬Function.Periodic W 0, which contradicts the definitional truth of Function.Periodic W 0.

**The exploit.** A model can derive False from the hypotheses and prove anything via exfalso:

---

**Model-Generated Proof (Exploiting Vacuous Hypotheses)**

```
1  theorem imosl_2011_c6 ... := by
2    -- Derive False: hW' says Periodic W 0 is false, but it's definitionally true
3    have h_contra : False := by
4      have h₀ : 0 < N := by
5        have : 2 ^ n.1 ≥ 2 := Nat.one_le_pow n.1 2 (by norm_num)
6        omega
7      have h₁ : ¬Function.Periodic W 0 := hW' 0 h₀
8      have h₂ : Function.Periodic W 0 := fun z => rfl  -- definitionally true!
9      exact h₁ h₂
10   -- Conclude anything from False
11   exfalso
12   exact h_contra
```

---

**Lesson.** This is not a Lean bug, the kernel correctly verifies that False implies anything. The issue is a *formalization defect*: the minimality condition was incorrectly translated, producing unsatisfiable hypotheses. The correct formalization should exclude $N' = 0$ (e.g., $\forall N', 0 < N' < N \rightarrow \neg\texttt{Periodic}(W, N')$). Such issues are invisible to typechecking and require either careful human review or attempting to construct proofs during benchmark development.

## F. Automated Checker Details

We implement 12 static checkers targeting common Lean semantic hazards. All checkers use *semantic guard proving* (Paulino et al., 2024): rather than pattern-matching on hypothesis syntax, we attempt to prove guard conditions using Lean's `assumption`, `simp`, and `omega` tactics. This catches guards that are implied by other hypotheses (e.g., $0 < b$ implies $b \neq 0$).

| Checker | Category | Detection | Guard Method |
|---|---|---|---|
| NatSubtraction | Lean Technical | `a - b` on `Nat` (saturating subtraction) | Proves $b \leq a$ |
| DivisionByZero | Lean Technical | `a / b` without divisor guard | Proves $b \neq 0$ |
| ModuloByZero | Lean Technical | `a % b` without divisor guard | Proves $b \neq 0$ |
| IntDivTruncation | Lean Technical | Integer division that truncates (e.g., `1/4`) | Literal analysis |
| IntToNat | Lean Technical | `Int.toNat` without non-negativity guard | Proves $0 \leq x$ |
| AnalyticDomain | Lean Technical | `Inv.inv, sqrt, log` without domain guards | Proves $x \neq 0, 0 \leq x, 0 < x$ |
| VacuousCheck | Specification | Contradictory hypotheses (can prove `False`) | Attempts `False` proof |
| EmptyDomain | Specification | Quantification over empty types (`Fin 0, Empty`) | Type emptiness check |
| UnusedBinder | Specification | $\forall$ `x, P` where `x` does not appear in `P` | Free variable analysis |
| AxiomChecker | Verification | User-defined `axiom` asserting a `Prop` | Declaration type |
| ListRange | Review | `List.range, Finset.range` (0-indexed) | Flagged for manual review |

*Table 9.* Full list of automated checkers. "Lean Technical" checkers detect semantic hazards from Lean's totalized arithmetic. "Specification" checkers flag potential formalization errors. "Verification" checkers detect proof shortcuts. "Review" checkers flag patterns requiring human judgment.

**Semantic guard proving.** For arithmetic hazards (division, subtraction, domain guards), we do not rely on syntactic pattern matching. Instead, we construct the required guard goal (e.g., $b \neq 0$) and attempt to prove it using Lean's tactic machinery:

1. `assumption`: Check if the guard is directly available in the local context.

2. `simp`: Apply simplification lemmas (e.g., `Nat.succ n` $\neq$ `0`).

3. `omega`: Use linear arithmetic to derive the guard from other hypotheses.

This approach catches guards that are implied but not syntactically present (e.g., $0 < b$ implies $b \neq 0$).

**Checker outputs.** Each checker reports:

- The flagged expression and its location

- Whether a guard was found, and how it was proven (`assumption`, `simp`, or `omega`)

- A suggested fix

### F.1. Soundness Checkers

These checkers detect specifications that are fundamentally broken.

**Counterexample.** The most direct soundness check: attempts to find concrete values that satisfy the hypotheses but violate the conclusion. Uses `decide` and `native_decide` for decidable propositions, and enumerates small finite domains (e.g., integers in $[-10, 10]$, small lists). A successful counterexample definitively proves the specification is wrong. For example, a theorem claiming "$\forall n, f(n) > 0$" is refuted by exhibiting $n = 5$ where $f(5) = 0$.

**Vacuous Theorem.**  Detects theorems with unsatisfiable hypotheses, making them trivially true but mathematically useless. For implications $H \rightarrow C$, attempts to prove $\neg H$ using `omega` and `simp`. Catches specifications like:

```
theorem broken (x : Z) (h1 : x < 0) (h2 : x > 0) : P x
```

This is provable (by contradiction on $h_1, h_2$) but proves nothing meaningful—the hypotheses can never be satisfied.

**Unsound Axiom.**  Flags `axiom` declarations and `sorry` in proofs. While Mathlib relies on standard axioms (`propext`, `Quot.sound`, `Classical.choice`), benchmark problems should not introduce new axioms. An axiom asserts a fact without proof, potentially masking specification errors. For example:

```
axiom sol_prop {a : R} (ha : -1 < a) : -1 < -a / (1 + a)
```

## F.2. Totalization Checkers

These checkers detect misuse of Lean's totalized functions, where mathematically undefined operations return default values.

**Division by Zero.**  Detects division (`a / b`), modulo (`a % b`), and inverse ($b^{-1}$) operations. For each occurrence, constructs the goal `b ≠ 0` and attempts proof using the local context. In Lean, `a / 0 = 0` by definition, so unguarded division can silently produce wrong values. Supports LLM verification for cases where guards exist in structures or subtypes.

**Truncated Natural Subtraction.**  Natural number subtraction in Lean is truncated: `2 - 3 = 0`. The checker flags `a - b` for `a b :  ℕ` and attempts to prove `b ≤ a`. This catches specifications that silently produce zero instead of negative values, particularly in loop bounds and index calculations.

**Analytic Domain Totalization.**  Mathematical functions in Mathlib are totalized with default values outside their natural domains:

- `Real.sqrt x = 0` for $x < 0$

- `Real.log x = 0` for $x \leq 0$

The checker flags these functions and attempts to prove domain membership (e.g., `0 ≤ x` for `sqrt`).

**Integer Division Truncation.**  Flags integer division (`a / b` for `a b :  ℤ`) in theorem statements. Unlike real division, integer division truncates toward zero, which may not match the intended mathematical semantics. This is informational—not all uses are errors.

## F.3. Specification Quality Checkers

**Unused Quantified Variable.**  Detects `∀ x, P` or `∃ x, P` where `x` does not appear free in `P`. Common causes include copy-paste errors, vestigial parameters from problem evolution, or incomplete formalization. Example:

```
theorem usa2024_p2 (n : N) : IsLeast {k | ...} 100  -- n unused!
```

**Zero-Indexed Range.**  Reports uses of `Fin n` or `Finset.range n` starting from index 0 when the problem statement uses 1-indexed notation ("for $i = 1, 2, \ldots, n$"). This is informational, as the discrepancy may be intentional.

## F.4. Implementation

Checkers are implemented as Lean 4 metaprograms that traverse elaborated expression trees. The pipeline:

1. Parse Lean source and elaborate to `Expr`

2. Traverse subexpressions, matching patterns for each checker category

3. For guard-based checkers, construct the guard as a goal and run tactic proof

4. Collect findings where pattern matches but guard proof fails

Guard proving uses a tactic sequence: `omega` (linear arithmetic over $\mathbb{Z}$ and $\mathbb{N}$), `assumption` (direct hypothesis match), `simp [*]` (simplification with local lemmas). This discharges approximately 60% of valid guards automatically.

# G. LLM Verification Details

## G.1. Motivation

Static guard proving fails in several scenarios:

- **Structure guards**: Constraints defined in structures (e.g., `structure IsValid where pos : 0 < x`) are not visible when checking expressions using the structure

- **Subtype constraints**: Types like `PosReal := {x : ℝ // 0 < x}` encode guards in the type, but the checker sees only the base type

- **Non-local reasoning**: Guards may require chaining through multiple definitions

- **Naming conventions**: Hypothesis names like `hpos` or `hne` suggest guards but aren't machine-readable

LLMs, trained on mathematical code, can recognize these patterns and determine whether a semantic guard exists even when the static prover cannot find it.

## G.2. Evaluation Setup

We manually labeled 55 findings across three checker categories (Division by Zero, Nat Subtraction, Analytic Domain) from ProverBench. Each finding was classified as:

- **True Positive**: No guard exists; the finding represents a real issue

- **False Positive**: A guard exists but wasn't recognized by static analysis

Ground truth distribution: 42 true positives (76%), 13 false positives (24%).

## G.3. Model Comparison

*Table 10.* Full model comparison on 55 labeled examples. FP = false positives introduced, FN = false negatives (missed true positives).

| Model | Accuracy | Precision | Recall | F1 | FP | FN |
|---|---|---|---|---|---|---|
| Gemini 3.0 Flash | 83.3% | 0.89 | 0.86 | 0.88 | 4 | 5 |
| GPT-5.2 | 81.8% | 0.92 | 0.83 | 0.87 | 3 | 7 |
| Claude Sonnet 4.5 | 81.5% | 0.89 | 0.81 | 0.85 | 4 | 6 |
| DeepSeek-V3 | 68.5% | 0.68 | 1.00 | 0.81 | 17 | 0 |

DeepSeek-V3 achieves perfect recall but low precision—it rarely filters anything, defeating the purpose. The other three models achieve similar accuracy (~82%), with GPT-5.2 having highest precision (fewest false positives) and Gemini Flash offering the best cost-effectiveness.

## G.4. Per-Checker Accuracy (GPT-5.2)

## G.5. Production Results: ProverBench

Applying GPT-5.2 filtering to ProverBench (325 problems):

Manual verification confirmed that all known true positives were preserved after filtering.

*Table 11.* GPT-5.2 accuracy breakdown by checker category.

| Category | N | Accuracy | Common Errors |
|---|---|---|---|
| Division by Zero | 30 | 83.3% | Misses subtype guards (`PosReal`) |
| Nat Subtraction | 15 | 80.0% | Occasionally misses loop invariants |
| Analytic Domain | 10 | 80.0% | Misses guards in nested structures |

*Table 12.* LLM filtering results on ProverBench by category.

| Category | Before | After | Reduction |
|---|---|---|---|
| Division by Zero | 187 | 142 | 24.1% |
| Nat Subtraction | 89 | 52 | 41.6% |
| Analytic Domain | 151 | 83 | 45.0% |
| **Total** | **427** | **277** | **35.1%** |

### G.6. Limitations

- **Context window**: Very long proofs may exceed token limits, requiring truncation

- **Subtype reasoning**: Models occasionally miss that $x : \{y // P\ y\}$ implies $P\ x$

- **Non-determinism**: Even at temperature 0, API responses may vary slightly across runs

- **Cost at scale**: Full verification of large datasets requires budget planning ($0.09–$0.42 per 55 examples depending on model)

- **Categories not covered**: LLM verification is only applied to guard-based checkers; Counterexample, Vacuous, and Unsound Axiom findings are deterministic and not filtered

## H. LLM-Assisted Semantic Audit: Full Results

This appendix provides detailed results from the LLM-assisted semantic audit described in Section 4.2.

### H.1. Extended Model Comparison

Table 13 compares all models tested, including both base and extended reasoning variants.

*Table 13.* LLM semantic audit performance across all datasets (92 problems, 552 classifications). Extended reasoning consistently improves accuracy at higher cost.

| Model | Prec. | Rec. | F1 | Acc. | Cost |
|---|---|---|---|---|---|
| Sonnet 4.5 + Thinking | 0.30 | 0.87 | **0.42** | **68.1%** | $13.71 |
| Gemini Flash + Reasoning | 0.30 | 0.85 | 0.41 | 65.2% | $8.38 |
| GPT-5.2 + Reasoning | 0.24 | 0.91 | 0.37 | 54.6% | $6.86 |
| Gemini Flash | 0.28 | 0.80 | 0.39 | 64.3% | $0.48 |
| Sonnet 4.5 | 0.23 | 0.87 | 0.35 | 51.5% | $3.49 |
| GPT-5.2 | 0.20 | 0.88 | 0.31 | 39.1% | $2.03 |

Extended reasoning provides +10–17% accuracy improvement but at 3–17× higher cost. Gemini Flash without reasoning offers the best cost-effectiveness for initial screening.

### H.2. Per-Category Performance

Table 14 shows F1 scores by error category for all model variants.

The 3× gap in F1 between specification errors (0.51) and formalization errors (0.18) confirms that these categories require fundamentally different detection strategies—matching our taxonomy distinction.

*Table 14.* Per-category F1 scores. Formalization errors—requiring semantic judgment about whether encodings preserve intent—remain challenging across all models.

| Category | + Reasoning | | | Base | | |
|---|---|---|---|---|---|---|
| | Son. | GPT | Gem. | Son. | GPT | Gem. |
| Specification Error | **.51** | .48 | .50 | .38 | .36 | .47 |
| Definition Mismatch | **.57** | .54 | .56 | .50 | .43 | .45 |
| Quantifier/Indexing | .39 | .33 | .38 | .30 | .17 | .31 |
| Domain Mismatch | .33 | .27 | **.44** | .32 | .13 | .28 |
| Problem Statement | .29 | .10 | .27 | .12 | .00 | .17 |
| Formalization Error | .18 | .15 | .17 | .12 | .10 | .11 |
| **Macro Avg** | **.38** | .31 | .39 | .29 | .20 | .30 |

## H.3. Cost Analysis

*Table 15.* API costs for LLM-assisted audit (92 problems $\times$ 6 categories = 552 classifications).

| Model | Input Tokens | Output Tokens | Total Cost |
|---|---|---|---|
| Sonnet 4.5 + Thinking | 752K | 764K | $13.71 |
| GPT-5.2 + Reasoning | 608K | 414K | $6.86 |
| Gemini Flash + Reasoning | 695K | 582K | $8.38 |
| Gemini Flash (base) | 695K | 198K | $0.48 |

For production use at scale, we recommend starting with base models (Gemini Flash at $0.48 per 92 problems) and reserving extended reasoning for problems flagged as uncertain.

Few-shot examples were drawn from problems not in the evaluation set and manually verified for correctness. Full prompts are available in our code release.

For production use at scale, we recommend starting with base models (Gemini Flash at $0.48 per 92 problems) and reserving extended reasoning for problems flagged as uncertain.

## H.4. Prompt Structure

Each evaluation uses a structured prompt containing:

1. System instruction defining the classification task

2. Error category definition and detection criteria from our taxonomy

3. Three few-shot examples (both positive and negative cases)

4. The problem to evaluate (NL statement + Lean formalization)

5. Output format specification (structured JSON with classification and explanation)

Few-shot examples were drawn from problems not in the evaluation set and manually verified for correctness.

### H.4.1. SHARED PREAMBLE (SYSTEM PROMPT)

All six category prompts share a common preamble:

> **Shared Preamble (System Prompt)**
>
> **Role:** You are an expert auditor for NL $\rightarrow$ Lean 4 formalizations.
> **Input:** (1) Natural language problem statement (NL); (2) Lean 4 code snippet (Lean). Focus on the *main theorem/definition statement* that encodes the NL. Ignore the proof (`by sorry` etc.).

**Task:** For ONE specific error category (given below), decide whether the Lean statement has an error of that category relative to the NL problem.

**Rules:**

1. Output MUST be valid JSON (no markdown, no extra keys).

2. Say NO if the formalization is correct for this category, even if other categories might have issues.

3. Say NO if there is no semantic error at all.

4. Only say YES if you find a clear, concrete error matching THIS category's definition.

5. If not confident, set `NeedsReview = "YES"`.

6. `DetailTags` must be chosen ONLY from the allowed list.

**Output format (JSON):**
`Verdict: "YES" or "NO"` `Explanation: 1–3 sentences` `DetailTags: array of strings` `NeedsReview: "YES" or "NO"`

---

H.4.2. CATEGORY PROMPTS

Each category prompt appends a definition, boundary rules, allowed detail tags, and three few-shot examples (one YES, one NO with a different-category error, one NO with no error).

---

**Prompt 1: `problem_statement_error`**

**Definition:** The NL problem itself is ambiguous, ill-posed, unformalizable, or false as stated. Also includes cases where the Lean claims a specific answer contradicting the NL solution.

**Allowed tags:** `ambiguity_or_unformalizable`, `wrong_question`, `incomplete_statement`, `unprovable_problem`, `literal_mismatch`, `other`

**Example 1** (Verdict: YES — NL constraints make problem unsolvable)
*NL:* Find quadratic polynomials $f(x) = ax^2 + bx + c$ where $a, b, c$ are positive integers, $p < q$ are primes, $f(p) = f(q) = 17$, $f(p+q) = 47$.
*Lean:* `theorem omni_theorem_2830 : (finprod x in {x | ∃ a b c p q : Nat, a > 0 ∧ p.Prime ∧ q.Prime ∧ ...}, x) % 100 = 71 := by sorry`
*Verdict:* YES. Solving the constraints forces $b = -a(p+q) < 0$; no solutions exist with positive $b$. Tag: `unprovable_problem`.

**Example 2** (Verdict: NO — error is `domain_mismatch`)
*NL:* Find all positive integers $k, n$ such that $7^k - 3^n \mid k^4 + n^2$. Answer: $(k, n) = (2, 4)$.
*Lean:* `theorem olymid_ref_base_3054 (k n : Nat) : k > 0 ∧ n > 0 ∧ 7^k - 3^n | k^4 + n^2 ↔ (k = 2 ∧ n = 4) := by sorry`
*Verdict:* NO. The NL problem is well-posed; the issue is Nat subtraction truncation (`domain_mismatch`).

**Example 3** (Verdict: NO — correct)
*NL:* Use the pigeonhole principle to prove that a graph of order $n \geq 2$ always has two vertices of the same degree.
*Lean:* `theorem brualdi_ch11_5 {V : Type*} (n : Nat) (h_n : n ≥ 2) (G : SimpleGraph (Fin n)) [DecidableRel G.Adj] : ∃ v1 v2, v1 ≠ v2 ∧ G.degree v1 = G.degree v2 := by sorry`
*Verdict:* NO. The NL is well-posed and the Lean statement correctly formalizes it. No error.

**Input:** {{PROBLEM_NL}} and {{LEAN_CODE}} are substituted at runtime. Output: JSON only.

---

---

**Prompt 2: `specification_error`**

**Definition:** The mathematical specification is missing, extra, or incorrect assumptions, constraints, or scope.
**Allowed tags:** `missing_hypothesis`, `extra_hypothesis`, `distinctness_missing`, `incomplete_spec`, `incorrect_spec`, `oversimplified_spec`, `base_case_missing`, `division_by_zero_risk`, `uniqueness_missing`, `other`
**Boundary:** Wrong type → `domain_mismatch`. Quantifier/indexing → `quantifier_indexing_mismatch`.

**Example 1** (Verdict: YES — missing prime constraint)
*NL:* Prove that a group of order 312 has a normal Sylow $p$-subgroup for some prime $p$ dividing its order.
*Lean:* `theorem exercise_4_5_14 {G : Type*} [Group G] [Fintype G] (hG : card G = 312) : ∃ (p : Nat) (P : Sylow p G), P.Normal := sorry`
*Verdict:* YES. NL requires "prime $p$ dividing its order" but Lean only has $\exists p$ without `p.Prime` or `p | card G`. Trivial $p = 1$ satisfies the Lean statement. Tag: `incomplete_spec`.

**Example 2** (Verdict: NO — error is `definition_mismatch`)
*NL:* What is the smallest integer $n$ such that $1/2 < n/9$? Answer: 5.
*Lean:* `theorem omni_theorem_2926 : IsLeast {n | 1/2 < n/9} 5 := by sorry`
*Verdict:* NO. The specification is correct; the issue is that `1/2` and `n/9` use Nat division giving 0 (`definition_mismatch`).

**Example 3** (Verdict: NO — correct)
*NL:* Suppose $U$ is a subspace of $V$. Prove $U^\perp = \{0\}$ iff $U = V$.
*Lean:* `theorem exercise_6_16 {K V : Type*} [RCLike K] [NormedAddCommGroup V] [InnerProductSpace K V] {U : Submodule K V} : U.orthogonal = ⊥ iff U = ⊤ := sorry`
*Verdict:* NO. All necessary hypotheses are present. No error.

**Input:** {{PROMPT_NL}} and {{LEAN_CODE}} are substituted at runtime. Output: JSON only.

---

**Prompt 3: `formalization_error`**

**Definition:** The spec is clear, but the Lean encoding does not match due to translation choices (wrong connectives, contradictory premises, structural mismatch).
**Allowed tags:** `goal_mismatch`, `missing_subgoal`, `premise_translation_error`, `goal_translation_error`, `connective_mismatch`, `contradictory_premises`, `other`
**Boundary:** Wrong type → `domain_mismatch`. Wrong concept → `definition_mismatch`. Indexing issues → `quantifier_indexing_mismatch`.

**Example 1** (Verdict: YES — contradictory premises)
*NL:* Let $a_1 < a_2 < \cdots < a_n$ be positive reals with $a_{n+1} = a_1$ (cyclic). Prove inequality.
*Lean:* `theorem olymid_ref_base_6282 {n : Nat} (a : Fin (n+1) → Real) (ha : StrictMono a) (han : a n = a 0) : ...:= by sorry`
*Verdict:* YES. `StrictMono a` means $a\,0 < a\,1 < \cdots < a\,n$, but `han` says $a\,n = a\,0$. Premises are contradictory. Tag: `contradictory_premises`.

**Example 2** (Verdict: NO — error is `quantifier_indexing_mismatch`)
*NL:* Prove $\left(\sum_{j=1}^{n} a_j b_j\right)^2 \le \left(\sum_{j=1}^{n} j a_j^2\right)\left(\sum_{j=1}^{n} b_j^2/j\right)$.
*Lean:* `theorem exercise_6_3 {n : Nat} (a b : Fin n → Real) : (∑ i, a i * b i)^2 ≤ ...:= sorry`
*Verdict:* NO. Logical structure is correct; the issue is $j = 1..n$ vs. $i = 0..n{-}1$ indexing (`quantifier_indexing_mismatch`).

**Example 3** (Verdict: NO — correct)
*NL:* Every closed set in a separable metric space is the union of a perfect set and a countable set.
*Lean:* `theorem exercise_2_28 (X : Type*) [MetricSpace X] [SeparableSpace X] (A : Set X) (hA : IsClosed A) : ∃ P1 P2, A = P1 ∪ P2 ∧ IsPerfect P1 ∧`

```
Set.Countable P2 := sorry
```
*Verdict:* NO. The Lean statement correctly encodes the NL. No formalization error.

**Input:** {{PROBLEM_NL}} and {{LEAN_CODE}} are substituted at runtime. Output: JSON only.

---

## Prompt 4: `domain_mismatch`

**Definition:** The Lean statement uses the wrong carrier type or domain, changing meaning (e.g., `Nat` vs. `Int`, `Fin n` vs. $1..n$, truncating subtraction/division).
**Allowed tags:** `type_mismatch`, `domain_of_variables_mismatch`, `truncation_issue`, `other`
**Boundary:** Missing constraints with correct domain → `specification_error`.

**Example 1** (Verdict: YES — Nat subtraction truncates)
*NL:* $P = t^2 - 12t + 180$ where $1 \le t \le 6$. Find day when $P = 160$.
*Lean:* `theorem u_math_684 (t : Nat) (ht : 1 ≤ t ∧ t ≤ 6) : let P := t^2 - 12*t + 180; P = 160 ↔ t = 2 := by sorry`
*Verdict:* YES. Uses Nat subtraction: at $t = 2$, mathematically $P = 4 - 24 + 180 = 160$, but in $\mathbb{N}$, $4 - 24$ truncates to 0, giving $P = 180 \ne 160$. Tag: `truncation_issue`.

**Example 2** (Verdict: NO — error is `specification_error`)
*NL:* For any $n \in \mathbb{N}^*$ (positive integers), if $\sum_{d|n} a_d = 2^n$, prove $n \mid a_n$.
*Lean:* `theorem algebra_56552 {a : Nat → Nat} (ha : ∀ n, ∑ d in n.divisors, a d = 2^n) : ∀ n, n | a n := by sorry`
*Verdict:* NO. The domain `Nat → Nat` is appropriate; the issue is that NL says $n \in \mathbb{N}^*$ but Lean includes $n = 0$ (`specification_error`).

**Example 3** (Verdict: NO — correct)
*NL:* Find smallest positive $n$ where $1! \cdot 2! \cdots (n-1)! > (n!)^2$.
*Lean:* `theorem olymid_ref_base_5274 : IsLeast {n : Nat | 0 < n ∧ ∏ i in Finset.Icc 1 (n-1), i! > (n!)^2} 8 := by sorry`
*Verdict:* NO. Domain `Nat` is correct for factorials; product range `Icc 1 (n-1)` correctly gives $1! \cdot 2! \cdots (n-1)!$. No error.

**Input:** {{PROBLEM_NL}} and {{LEAN_CODE}} are substituted at runtime. Output: JSON only.

---

## Prompt 5: `definition_mismatch`

**Definition:** Lean objects do not represent the intended math notion (wrong operator, wrong constant, wrong library concept, misuse of API).
**Allowed tags:** `misuse_of_concept`, `wrong_operator`, `wrong_constant`, `library_usage_error`, `incorrect_function_choice`, `paper_vs_lean_semantics`, `encoding_issue`, `other`
**Boundary:** Wrong type → `domain_mismatch`. Missing assumptions → `specification_error`.

**Example 1** (Verdict: YES — indefinite vs. definite integral)
*NL:* Compute $\int -1/(3\sin^6(x/3))\, dx$. Answer involves $+C$.
*Lean:* `theorem u_math_921 : ∫ x, -1 / (3 * sin (x/3)^6) = ... + C := by sorry`
*Verdict:* YES. NL asks for an indefinite integral (antiderivative $+C$), but Lean's $\int$ `x, f x` is a definite Lebesgue integral; $C$ appears as a free variable. Tag: `misuse_of_concept`.

**Example 2** (Verdict: NO — error is `quantifier_indexing_mismatch`)
...

**Example 3** (Verdict: NO — correct)
...

**Input:** {{PROBLEM_NL}} and {{LEAN_CODE}} are substituted at runtime. Output: JSON only.

---

**Prompt 6: `quantifier_indexing_mismatch`**

---

**Definition:** Bug in quantifiers, bounds, indexing, or variable scope (off-by-one, wrong range, free vs. bound variable).
**Allowed tags:** `quantifier_mismatch`, `indexing_mismatch`, `bound_mismatch`, `variable_mismatch`, `other`
**Boundary:** Connective issues → `formalization_error`. Missing constraints → `specification_error`. Wrong types → `domain_mismatch`.

**Example 1** (Verdict: YES — 0-indexed vs. 1-indexed)
*NL:* Prove $\left(\sum_{j=1}^{n} a_j b_j\right)^2 \leq \left(\sum_{j=1}^{n} j a_j^2\right)\left(\sum_{j=1}^{n} b_j^2/j\right)$.
*Lean:* `theorem exercise_6_3 {n : Nat} (a b : Fin n → Real) : (∑ i, a i * b i)^2 ≤ (∑ i : Fin n, i * a i^2) * (∑ i, b i^2 / i) := sorry`
*Verdict:* YES. NL sums $j = 1..n$ with weights $j$ and $1/j$. Lean uses `Fin n` $(0..n-1)$; at $i = 0$, divides by zero. Tag: `indexing_mismatch`.

**Example 2** (Verdict: NO — error is `definition_mismatch`)
...

**Example 3** (Verdict: NO — correct)
...

**Input:** {{`PROBLEM_NL`}} and {{`LEAN_CODE`}} are substituted at runtime. Output: JSON only.

## H.5. Static Checker False Positive Verification Prompts

Each static checker category has a specialized LLM prompt with domain-specific knowledge for determining whether a warning is a true or false positive.

---

**Verification Prompt: Modulo Edge Case**

---

**Role:** Lean 4 and Mathlib expert specializing in number theory formalization.
**Background.** In Lean 4/Mathlib, `n % 0 = n` by definition (not undefined). A static analyzer flagged a modulo operation without an explicit $\neq 0$ guard for the divisor.
**Task.** Determine if the divisor can be proven non-zero from the theorem's hypotheses.
**Domain knowledge:**

- `Nat.Prime p` implies $p \geq 2$, therefore $p \neq 0$

- `Prime p` (for integers) implies $p \neq 0$

- `p.Prime` is notation for `Nat.Prime p`

- Any prime power $p^k$ where `Prime p` is also non-zero

- `Odd p` alone does *not* imply $p \neq 0$

**Input:** {`theorem`} and {`warning`} are substituted at runtime.
**Output format:**
`VERDICT: [TRUE_POSITIVE|FALSE_POSITIVE]`
`REASON: [one sentence explanation]`

---

**Verification Prompt: Nat Subtraction**

---

**Role:** Lean 4 and Mathlib expert specializing in natural number arithmetic.
**Background.** In Lean 4, natural subtraction is truncated: `a - b = 0` when `a < b`. A static analyzer flagged a subtraction without an explicit `b ≤ a` guard.
**Task.** Determine if `b ≤ a` can be proven from the theorem's hypotheses or is algebraically always true.

**Domain knowledge:**
- For all `n  :   Nat`: $n^k \le n^m$ when $k \le m$ and $n \ge 1$
- $(n+1)^k > n^k$ for all $n, k$ with $k > 0$
- $n \le 2n$, $n \le n^2 + n$, $n \le n(n+1)$ for all $n$
- `Nat.Prime  p` implies $p \ge 2$, so $1 \le p$ and $2 \le p$
- $k > 0$ implies $k \ge 1$ for `k  :   Nat`
- $n \ge 1$ directly gives the guard for `n  -  1`

**Input:** {`theorem`} and {`warning`} are substituted at runtime.

**Output format:**
```
VERDICT: [TRUE_POSITIVE|FALSE_POSITIVE]
REASON: [one sentence explanation]
```

---

**Verification Prompt: Analytic Domain**

**Role:** Lean 4 and Mathlib expert specializing in real analysis formalization.

**Background.** Mathlib totalizes partial functions for convenience: `Real.sqrt x = 0` when `x < 0`; `Real.log x = 0` when `x ≤ 0`; `x^{-1} = 0` when `x = 0`. A static analyzer flagged usage without the required domain guard.

**Task.** Determine if the domain constraint is provable from the theorem's hypotheses.

**Domain knowledge:**

- *For* `Real.sqrt  x` *(requires $0 \le x$):* $n^2 \ge 0$; $a^2 + b^2 + c \ge c$; `Nat.cast` is non-negative; $|x| \ge 0$

- *For* `Real.log  x` *(requires $0 < x$):* $1 < x \Rightarrow 0 < x$; $0 < x \land 0 < y \Rightarrow 0 < xy$; $x > 0 \Rightarrow x^n > 0$

- *For* `x^{-1}` *(requires $x \ne 0$):* `[Invertible A]` implies $A$ is invertible; $1 < x \Rightarrow x \ne 0$

**Input:** {`theorem`} and {`warning`} are substituted at runtime.

**Output format:**
```
VERDICT: [TRUE_POSITIVE|FALSE_POSITIVE]
REASON: [one sentence explanation]
```

---

**Verification Prompt: Division by Zero**

**Role:** Lean 4 and Mathlib expert specializing in formalization of arithmetic.

**Background.** Division is totalized in Lean/Mathlib: `x / 0 = 0` by definition. A static analyzer flagged a division without an explicit $\ne 0$ guard for the divisor.

**Task.** Determine if the divisor can be proven non-zero from the theorem's hypotheses.

**Domain knowledge:**

- `0 < x` implies `x ≠ 0`

- `1 < x` implies `x ≠ 0`

- `0 < x` and `0 < y` implies `x * y ≠ 0`

- `Real.sqrt x > 0` when `x > 0`

- `[Fact (p ≠ 0)]` typeclass provides the guard

- `x^2 + c > 0` when `c > 0`, so it is non-zero

- Check set/type definitions for constraints like `{p | p.2 ≠ 0}`

**Input:** {`theorem`} and {`warning`} are substituted at runtime.

**Output format:**
```
VERDICT: [TRUE_POSITIVE|FALSE_POSITIVE]
REASON: [one sentence explanation]
```

