# OpenReview forum: "Faults in Our Formal Benchmarking: Dataset Defects and Evaluation Failures in Lean Theorem Proving"
_ICML.cc/2026/Conference — ICML 2026 regular_

### Official Review · Reviewer_Tcmw · 2026-02-21

**Soundness:** 3
**Presentation:** 4
**Significance:** 3
**Originality:** 2
**Overall Recommendation:** 5
**Confidence:** 1

**Summary:**

The paper challenges the assumption that Lean-based benchmarks are sound because proofs are checked by a machine. They show why this assumption can be misleading. The authors audit some widely used benchmarks and find defects in each of them. The paper analyzes the full benchmarking pipeline, from informal problem --> Lean statement --> model output --> Lean kernel --> reported metric, and the key insight is that the "trust boundary" is only at the Lean kernel check: everything before and after can have defects. The paper provides more than a few concrete examples of such dataset defects. Further, they suggest a fault taxonomy: based on five popular benchmarks the authors provide a taxonomy of faults into fidelity issues, evaluation loopholes and maintenance decay (and their subcategories). To address these faults, they implement automated checkers of formal math datasets. To complete these, they suggest standards for creating trustworthy dataset.

**Compliance With Llm Reviewing Policy:**

Affirmed.

**Final Justification:**

I view this as an important and timely paper for the formal theorem proving community. My main concerns were about the lack of quantitative evidence for how benchmark defects affect reported scores, the role of the LLM-assisted semantic audit, and the broader relevance beyond Lean. After the rebuttal, these concerns were addressed, and in particular the added quantitative results make the empirical motivation substantially stronger. Overall, the rebuttal strengthened the paper, and I support acceptance.

**Key Questions For Authors:**

1) Can you quantify (or estimate) the amount of problems per benchmark which are affected by each of the fault categories, and how reported scores deviate from the correct ones?

2) Some of the issues are Lean-specific, how much of the taxonomy is relevant to other proof assistants? Are there adaptations that would be required that you can already identify?

**Limitations:**

yes

**Strengths And Weaknesses:**

Strengths

(-) The world of Theorem proving gains much attention these days, and the reliability of Lean benchmarks is becoming increasingly important. This work has potentially high impact.

(-) The benchmarking pipeline in Figure 1 is a clear conceptual contribution, of independent interest.

(-) The concrete examples that the paper provides are practically helpful and may be illuminating for people in this field.

(-) the recommendations in section 6 are all highly actionable.


Weaknesses

(-) I didn't find a quantitive estimate of how much the scores are affected by the defects of the dataset. Indeed it is convicing that flawed datasets are bad practice and should align with high standards, however measuring how much these defects actually change the benchmark scores would substantially strengthen the empirical case.

(-) I have some reservations about the "LLM-Assisted Semantic Audit" and I'm afraid that presenting it in the main body at this point may weaken your result.

(-) The paper only considers Lean but since the concept is also relevant to, say, Coq benchmarks, exploring another case study can further strengthen the arguments and maybe even be able to generalize some of the concepts further.

(-) in terms of presentation: it's not too central but it'll be good to improve the quality of figure 1.

---

> ### Author Rebuttal · Authors · 2026-03-31
>
> Thank you for the encouraging review. We are especially glad the trust-boundary framing, concrete examples, and release standards came through as intended. We address each point below.
>
> **Quantitative estimate of score impact.** We had run our static checkers corpus-wide but failed to include the aggregate results in the submission, which was an oversight we correct here. Across all five benchmarks and their forks (13 variants, ~10,000 problems), we surfaced 4,833 findings including 399 proven issues (counterexamples, unsound axioms, vacuous theorems). Per-benchmark breakdown:
>
> Our *static checkers* (Lean 4 metaprograms) were run on every problem across all five benchmarks and their forks (13 variants). Corpus-wide results:
>
> | Benchmark | Problems | Total Findings | Proven Issues |
> |---|---|---|---|
> | FormalMath (all) | 5,560 | 3,250 | 141 |
> | ProverBench | 325 | 370 | 208 |
> | miniF2F (6 forks) | ~490 each | 74–193 per fork | 0–13 per fork |
> | ProofNet (2 versions) | 371 each | 67–82 per version | 1–5 per version |
> | CombiBench | 100 | 78 | 1 |
> | **Total** | **~10,000** | **4,833** | **399** |
>
> Total numbers across benchmarks:
> | Benchmark | Problems | Findings | Proven | Div/0 | Nat Sub | Unsound Axiom | Counterex. | Vacuous |
> |---|---|---|---|---|---|---|---|---|
> | formalmath@all | 5560 | 3250 | 141 | 1127 | 582 | 0 | 55 | 1 |
> | proverbenchmath | 325 | 370 | 208 | 81 | 7 | 199 | 5 | 0 |
> | minif2f@v2c | 244 | 193 | 8 | 45 | 28 | 0 | 3 | 0 |
> | proofnet@deepseek | 371 | 82 | 5 | 13 | 8 | 0 | 5 | 0 |
> | minif2f@yangky11-early | 244 | 79 | 13 | 14 | 19 | 0 | 5 | 1 |
> | combibench@hf | 100 | 78 | 1 | 1 | 27 | 0 | 1 | 0 |
> | minif2f@justincasher | 244 | 74 | 0 | 25 | 13 | 0 | 0 | 0 |
> | proofnet@sharp | 371 | 67 | 1 | 11 | 6 | 0 | 1 | 0 |
> | minif2f@ai-mo-test | 244 | 37 | 1 | 17 | 6 | 0 | 0 | 0 |
>
> To measure how defects change reported scores, we provide two pieces of evidence with complementary error patterns:
>
> First, we selected 20 problems with proven issues across 4 datasets, manually corrected them, and evaluated provers on both versions. Both models solved 0/20 originals (the flawed specifications were unprovable); on corrected versions, DeepSeek-Prover-V2-7B solved 3/20 and Kimina-Prover-8B solved 2/20. These problems silently deflate scores by adding impossible items to the denominator.
>
> Second, Poiroux et al. independently corrected 118/371 ProofNet problems (32%), where the dominant pattern is the opposite: weaker specifications make problems artificially easier. On the 186-problem test split, pass rates dropped on corrected versions: Claude Opus 4.5 from 22.0% to 12.4% (44% relative), DeepSeek-Prover-V2-7B from 9.7% to 6.5%, Kimina-Prover-8B from 9.7% to 7.0%.
>
> These two patterns (unprovable statements deflating scores, weakened statements inflating them) coexist in the same benchmarks and can partially cancel in aggregate, making headline numbers misleading in both directions.
>
> **LLM-Assisted Semantic Audit.** We agree with this reservation. The LLM audit is a recall-oriented triage tool for error categories that static analysis cannot decide (missing hypotheses, definition mismatches), not a standalone validator. We will reframe its presentation in the camera-ready to make this role explicit: the static checkers are the primary scalable contribution, and the LLM component is an exploratory complement requiring human filtering.
>
> **Other proof assistants**. The taxonomy transfers, as the faults arise from the human-to-formal-system translation, and not from Lean's specific type theory. These affect any formal verification benchmark. For example, Rocq benchmarks face analogous issues: totalization of arithmetic in its standard library, changes in tactic behavior across versions (e.g., omega replacing romega), and specification drift as libraries like MathComp evolve. The concrete checkers are Lean-specific (Lean 4 metaprogramming, mathlib APIs), but the taxonomy, release standards, and audit methodology apply broadly. We will add a transferability discussion in the camera-ready.
>
>
> **Figure 1.** Will be improved for the final version.
>
> We are grateful for the reviewer's support and would welcome any further suggestions for strengthening the camera-ready.

---

> > ### Author Rebuttal · Reviewer_Tcmw · 2026-04-02
> >
> > Thank you for the thoughtful rebuttal. My concerns are fully resolved -- the added quantitative results on benchmark defects, together with the discussion of their impact on prover scores, make the empirical motivation much clearer and stronger. I encourage you to add these to your revision.
> >
> > I also appreciate the clarification that the LLM-assisted semantic audit is an exploratory complement to the static checkers rather than a primary validation mechanism, as well as the discussion of transferability to other proof assistants. Overall, the rebuttal clarified the contribution and addressed my main concerns.

---

### Official Review · Reviewer_NSRW · 2026-03-05

**Soundness:** 3
**Presentation:** 3
**Significance:** 3
**Originality:** 2
**Overall Recommendation:** 4
**Confidence:** 3

**Summary:**

The paper investigates the common errors that persist across numerous Lean datasets. The authors discuss how semantic drift, Lean-related bugs and other factors affect current formal math provers. Moreover, authors propose a fault taxonomy that aims to minimize these issues and detect them prior to dataset releases.

**Compliance With Llm Reviewing Policy:**

Affirmed.

**Final Justification:**

The rebuttal addressed the majority of my concerns, and I have increased the score to 'Weak Accept.' I find the tone and some of the justifications for the results lacking, but if the authors implement all promised changes, I do not have any major concerns that would warrant rejecting this work.

**Key Questions For Authors:**

- How the proposed taxonomy performs on the original Lean4 version of the miniF2F? Could you identify mistakes in that dataset pointed out by other researchers?
- To what extent do introduced corrections affect the theorem provers? Can the current generation of models prove altered statements? Perhaps some ablations are needed to establish the effect of introduced changes.

**Limitations:**

Yes, authors adequately discussed limitations of the work.

**Strengths And Weaknesses:**

## **Strengths**
- **Strong message**. I believe the strongest part of the paper is the message to the community. In recent years, benchmarking in Lean has become extremely difficult due to technical and non-technical issues. Many benchmarks are unrunnable due to version conflicts, and models become outdated due to under-the-hood tactic changes. I believe the problems raised by this work are highly relevant to the formal math community.

- **The work details common Lean benchmarking pitfalls**. The paper outlines and classifies common benchmarking issues with clarifying examples. The terminology in this work is straightforward and easy to digest. I believe this is an important strength, since clarity is crucial when communicating the shortcomings of benchmarking.

- **The proposed taxonomy is straightforward**. The outlined method that combines automatic and LLM-assisted checkers seems reasonable. Although applying an LLM for semantic drift verification has its disadvantages, the authors communicate and experimentally ablate the LLM component with sufficient clarity.

With that being said, there are certain weaknesses that I would like the authors to address.

---

## **Weaknesses**

- **The work identifies pitfalls but does not patch the benchmarks.** Correct me if I am wrong, but the authors do not claim to release the evaluated datasets as "platinum benchmarks." If this is the case, the impact of the paper is limited to the proposed taxonomy only. Could the authors please clarify this?
- **Limited numerical section.** The authors identify critical downfalls of the benchmarks but do not answer the most essential question when it comes to evaluation: how do these mistakes, changes, and bugs affect the provers? I could not find the post-correction accuracy of the provers. It seems like a natural extension of the paper to investigate the extent to which these issues affect the reliability of the current generation of provers. Moreover, miniF2F-Lean4 (which is by far more popular than the Lean3 version) is omitted from the paper's discussion. Why is this? It seems like an oversight to not include miniF2F-Lean4.
- **Additional discussion of related work is needed.** The paper does not mention recent attempts at fixing the identified bugs and mistakes in the miniF2F-Lean4 benchmark. Numerous researchers have identified misalignments in miniF2F and released "fixed" versions [1, 2, 3]. Furthermore, [1] released a fully patched version of miniF2F (miniF2F-v2) with a similar core message about benchmarking pitfalls (semantic drift, unprovable problems, incorrect variable types, etc.). Since the messages of these papers overlap, I believe the authors should expand their discussion beyond merely stating that these versions exist. Moreover, it would be interesting to see the effectiveness of the taxonomy on benchmarks that already have a "ground truth" dataset with human-identified errors, i.e. comparing the taxonomy-identified statements against the human-identified ones.

---

[1] Ospanov et al. (2025). miniF2F-Lean Revisited: Reviewing Limitations and Charting a Path Forward. NeurIPS 2025.

[2] Wang et al. (2025) Kimina-Prover Preview: Towards large formal reasoning models with reinforcement learning. arXiv.

[3] Liu et al. (2025) Atlas: Autoformalizing theorems through lifting, augmentation, and synthesis of data. NeurIPS 2025.

---

> ### Author Rebuttal · Authors · 2026-03-31
>
> Thank you for the constructive review. We are glad the message, clarity, and taxonomy came through as intended. We address each concern below.
>
> **Patching the benchmarks.** You are correct that we do not release corrected "platinum" versions, and we agree this would be valuable. However, producing verified corrections at scale is substantially more expensive than it may appear, and the benchmarks we audit were not carelessly assembled.
>
> FormalMath recruited 12 IMO-medalist-level experts solely to check semantic equivalence between NL and Lean (not to fix anything) at $6.89 per statement, over 22 days, with a preservation rate of only 72%. ProofNet included human review at the time of creation, yet ProofNet# later identified errors in 118 of 371 problems (32%). MiniF2F has been publicly available for four years and has undergone multiple independent community repair efforts, yet Ospanov et al. (2025) found issues in 50% of statements in v1. These results illustrate that even careful human verification routinely misses subtle formalization errors.
>
> Extrapolating to our five-benchmark suite (6,844 problems), we estimate a platinum-standard correction would require: (1) source-math verification by 2-3 PhD-level mathematicians per problem (about \\$140K), (2) NL-to-Lean fidelity checking at US academic rates (about \\$140-210K), (3) Lean encoding repair for the 25-30% of problems likely to need fixes, by specialized Lean experts (~\\$125-250K). The total is conservatively \\$300-500K before ongoing maintenance, which is itself non-trivial as Lean and mathlib evolve on a monthly release cadence. This is a multi-team research program, not a single-paper contribution.
>
> Our automated static checkers provide the scalable first pass that makes future expert-driven correction tractable: they screen all 6,844 (original count not including forks) problems at near-zero marginal cost, flagging the 10-30% most likely to contain defects and directing scarce expert attention where it is most needed.
>
> **Prover impact.** We provide two pieces of evidence showing that defects materially affect scores.
>
> First, we selected 20 problems with proven issues (counterexamples or vacuous hypotheses) across 4 datasets, manually corrected them, and evaluated provers on both versions. On the originals, both models solved 0/20: the flawed specifications rendered these statements unprovable. On the corrected versions, DeepSeek-Prover-V2-7B solved 3/20 and Kimina-Prover-8B solved 2/20. The originals were not "easier with bugs"; they were unprovable, yet their presence silently deflates scores by adding impossible problems to the denominator.
>
> Second, at a larger scale: Poiroux et al. independently corrected 118/371 ProofNet problems (32%). Here the pattern differs: ProofNet often encodes weaker versions of problems, making them artificially easier. On the 186-problem test split, pass rates dropped on corrected versions: Claude Opus 4.5 from 22.0% to 12.4% (44% relative drop), DeepSeek-Prover-V2-7B from 9.7% to 6.5%, Kimina-Prover-8B from 9.7% to 7.0%.
>
> These two patterns (unprovable statements deflating scores, weakened statements inflating them) can coexist in the same benchmark and partially cancel in aggregate, making headline numbers misleading in both directions.
>
> **miniF2F-Lean4 and related work.** We ran our static checkers on 6 versions miniF2F-Lean4, including Ospanov et al.'s corrected v2. On v2c (complete split): 193 findings, 8 proven issues; on v2s (statement-fixed split): 181 findings, 8 proven issues. We found 3 counterexamples and 5 truncation errors in their corrected dataset that persisted through expert repair, confirming that human review and automated static analysis catch different classes of defects. We discuss various versions of miniF2F-Lean4 in Section 4.3, and will add more details about how the versions fix minif2f in the camera ready version.
>
> Regarding ground-truth validation: the reviewer suggests comparing our taxonomy-identified errors against human-identified ones, treating miniF2F-v2 as ground truth. However, since our checkers found 3 counterexamples and 5 truncation errors in v2 itself, we cannot treat it as a reliable gold standard for this comparison. We are currently annotating Harmonic's miniF2F version (which, to our knowledge, contains no issues flagged by our checkers) and will aim to provide the cross-comparison the reviewer requests by the end of the discussion period.
>
> We will expand the related work to engage substantively with Ospanov et al. (2025), Wang et al. (2025), and Liu et al. (2025).
>
> We hope this addresses the main concerns, and thank the reviewer for their time and effort to review the paper.

---

> > ### Author Rebuttal · Reviewer_NSRW · 2026-04-01
> >
> > I thank the authors for their thorough rebuttal and further clarifications. Although the cost estimations presented for the platinum benchmarks appear overestimated, I believe the work carries an important message. I am willing to increase my score by one.
> >
> > Human review is inherently prone to errors due to negligence, subjectivity, and other factors. Going forward, it would be best if the authors do not diminish the efforts of researchers who have attempted to fix and improve upon previous dataset versions. In particular, claiming that the published collection of the latest error fixes (including inconsistencies found in miniF2F-v2) is not sufficient to run a comparative analysis borders on disrespect toward the authors of previously published works. I believe that in a paper like this, authors should acknowledge those who came before and treat the subject with care. At the current stage of research, _human review remains the gold standard_, whether we like it or not. Maintaining a balanced perspective on the limitations of both human review and automated fixes would strengthen the paper’s objectivity.
> >
> > It would be interesting to see how the authors tackle the problem of platinum benchmarks in Lean beyond mere identification. I wish the authors the best of luck and look forward to seeing what comes next.

---

> > > ### Author Response · Authors · 2026-04-06
> > >
> > > Thank you for the thoughtful feedback on tone. We agree that prior human repair efforts should be treated with care. We now frame them as high-quality reference points for comparative analysis, and we agree that human review remains the gold standard for final semantic adjudication.
> > >
> > > Following your suggestion, we performed a cross-version audit of the 244-problem test split of the original Lean4 yangky11-early miniF2F port [An early commit pinned to Lean 4.3.0]. We compared it against prior repaired variants, especially miniF2F-v2c and Harmonic’s Lean4 re-formalization, and then manually adjudicated disagreements under our taxonomy. In addition to manual annotation, we used three independent model-assisted preliminary labeling passes; these were unanimous on 236/244 items and, together, yielded 41 confirmed problematic statements.
> > >
> > > The main result is that our automated audit is complementary to human expert translation. Our automated checkers help by enabling human experts to focus on potentially problematic translations and rely on human expertise to adjudicate. Many of the confirmed issues are already addressed in prior repaired variants, which validates that the taxonomy recovers human-identified errors. The remaining divergences are concentrated in Lean4-port-specific formulation issues and mechanically checkable hazards (e.g., truncation and precedence problems) that are easy to miss in manual review but amenable to automated checking.
> > >
> > > This is also how we interpret miniF2F-v2c: A strong human reference point that usefully overlaps with our audit while still leaving room for automated checks to catch a different class of issues. In that sense, human review and automated auditing are complementary,
> > > and supports the paper’s main claim.
> > >
> > > As a secondary check, our miniF2F semantic-audit experiment again found that model-based semantic review is helpful for recall-oriented triage, but it is still not a substitute for human adjudication.
> > >
> > > We will release the problematic statements, with taxonomy labels, to support semantic audit evaluation and gold labels from the reference datasets, which could also serve as an autoformalization evaluation dataset.
> > >
> > > We thank the reviewer for their time and effort, and will make appropriate changes in the camera ready version.

---

### Official Review · Reviewer_21Xw · 2026-03-09

**Soundness:** 2
**Presentation:** 2
**Significance:** 3
**Originality:** 3
**Overall Recommendation:** 4
**Confidence:** 3

**Summary:**

The study investigates the fidelity issue when translating natural language problems descriptions to formal Lean statements. The authors highlight that Lean theorem prover only verifies that all open goals are closed given a formal statement, however it does not verify whether the formal statement faithfully encodes the problem. They audit existing datasets. such as miniF2F, ProofNet, FormalMath, CombiBench, ProverBench, and find missing hypotheses, simplifications, incomplete and incorrect translations. To address the issue and improve existing benchmarks, they propose a fault taxonomy and tools for automated evaluation. The experimental result show that llm based semantic evaluation can detect specification errors and definition mismatches but struggle with formalization errors.

**Compliance With Llm Reviewing Policy:**

Affirmed.

**Final Justification:**

The authors have addressed the concerns raised during the review and provided access to the dataset. I would have appreciated more time during the review to evaluate the resulting datasets. After checking a few details in the uploaded repository, I raised my overall score and believe that the work provides a useful contribution to the field that others may build on.

**Key Questions For Authors:**

- Section 4.2 missing reference for DeepSeek-Prover-V2
- Who annotated the 55 examples for LLM based semantic equivalence checking? And is that dataset available?
- Why do you select a subset of the problems for automated auditing in Section 5.2? How was this dataset sampled (number of problems per benchmarks is not equal)?
- The semantic LLM audit performance results are quite low. How can semantic fidelity evaluation be improved?

**Limitations:**

yes

**Strengths And Weaknesses:**

## Strengths

- The study is well motivated and identifies, evaluates and corrects issues in existing autoformalisation and theorem proving benchmark datasets.
- The authors provide feedback and guidelines for benchmark improvement.
- Evaluate static analysis tool and LLM based semantic analysis for automating benchmark audits.

---

## Weaknesses

- The study is not well embedded in existing literature evaluating fidelity in autoformalisation and is missing several references, including recent references listed below:
    - [Ospanovet et al., NEURIPS 2025](https://openreview.net/forum?id=KtaHv0YUyh)
    - [Liu et al., ICLR 2025](https://openreview.net/forum?id=hUb2At2DsQ)
    - [Cabral et al., ICLR 2026](https://openreview.net/forum?id=s9t2FJVsBH)
- The information about the dataset audit and. evaluation strategy is missing important details. Who conducted the audit? Who validated the findings?
- Sections 1-4 include multiple repetitions or errors and error types. I suggest writing a comprehensive section explaining all relevant errors and types and using the additional space for further analysis and details in the experiments.
- Do not provide the corrected benchmarks and resources used for evaluation, including the few-shot examples and prompts for LLM evaluation.

---

> ### Author Rebuttal · Authors · 2026-03-31
>
> Thank you for the detailed review. We agree that (1) the paper should be better embedded in recent literature, and (2) the annotation and sampling protocol should be stated much more explicitly. We address other concerns below.
>
> **Patching the benchmarks.** You are correct that we do not release corrected versions. We want to emphasize that producing verified corrections is substantially harder than it may appear. The benchmarks we audit were not carelessly assembled. Each invested in human verification, and each still contains the defects we document:
>
> - FormalMath recruited 12 IMO-medalist-level experts solely to check semantic equivalence (not to fix anything), at $6.89/statement over 22 days, yet still achieved only 72% preservation.
> - ProofNet included human review at creation time. ProofNet# later found errors in 118 of 371 problems (32%).
> - MiniF2F has been publicly available for four years, has been scrutinized by dozens of researchers, and has undergone at least five independent repair efforts. Ospanov et al. (2025) still found issues in over 50% of v1 statements.
>
> This is the central point: if datasets built by domain experts, reviewed by IMO medalists, and maintained by active communities still contain 30-50% error rates, then producing a "platinum" correction is not a matter of running one more pass. It requires multi-annotator consensus (2-3 mathematicians per problem for source verification, 2 Lean experts per problem for encoding correctness), at a cost we estimate conservatively at $300-500K across our five-benchmark suite (6,844 problems (not fixing forks)) before ongoing maintenance as Lean and mathlib evolve monthly.
>
> **Missing references.** Thank you for the pointers. We will incorporate Ospanov et al. (NeurIPS 2025), Liu et al. (ICLR 2025), and Cabral et al. (ICLR 2026) into the related work and draw explicit comparisons.
>
> **Who conducted the audit? Who validated the findings?** The paper contains two distinct evaluation components that we will separate more clearly in writing.
>
> The *static checkers* (Section 5.1) are deterministic Lean 4 metaprograms with unit and integration tests, run corpus-wide across all five benchmarks (~10,000 problems including forks), and surface 4,833 findings and 399 proven issues. Findings are validated by construction: each checker attempts to discharge a guard condition using Lean's own tactic machinery and reports only when the guard cannot be proven.
>
> The *55 labeled items* (Table 4) constitute a ProverBench validation set for LLM false-positive filtering. We (the authors) manually labeled these.
>
> The *92 semantic-audit items* (Tables 5-6) are a curated challenge set, not a random sample. They were sourced from: (a) CombiBench GitHub issues (community-identified), (b) ProofNet vs. ProofNet# diffs, and (c) manual annotation of FormalMath and ProverBench by the authors. This is the complete set with ground-truth labels. No subsampling was performed. Unequal per-benchmark counts reflect where errors had been documented.
>
> **Repetition in Sections 1-4.** We agree. We will consolidate the presentation, freeing space for experimental detail.
>
> **Semantic LLM audit performance.** Specification errors and definition mismatches are more detectable (F1 0.48-0.57) than formalization errors (F1 0.15-0.18), because the former involve explicit differences in the objects translated, while the latter require deeper semantic judgment. Regarding the reviewer's question about how to improve, we see three directions. First, datasets with ground truth would enhance the ability to hill climb this benchmark. Additionally, we hope that releasing our labeled dataset (92 items with ground-truth error annotations) provides a benchmark for this task where none currently exists. Second, giving models access to Lean environment explorers (tools that can inspect how mathematical objects are defined in mathlib, query type signatures, and check definitional equivalences) would address a key bottleneck that models currently lack the context to judge whether a Lean encoding faithfully captures mathematical intent. Third, as reasoning models improve, we expect the semantic gap to narrow, particularly for formalization errors.
>
> **Minor points:** DeepSeek-Prover-V2 citation will be added in Section 4.2. Few-shot prompts are in Appendix H.4; we will update the paper in the camera-ready version to surface this fact better. With the camera-ready, we will release: (a) static checker code, (b) labeled evaluation sets, (c) all LLM prompts, (d) full checker outputs.
>
> We hope this addresses the main concerns and thank the reviewer again for their feedback on improving the paper.

---

> > ### Author Rebuttal · Reviewer_21Xw · 2026-04-01
> >
> > Thank you for your detailed response addressing my comments and concerns. I believe that the discussed changes improve the presentation of the manuscript. A major concern regarding a core contribution of the paper remains, as the application of the proposed tools and verification pipeline to existing benchmarks cannot be verified without access to the data resources.

---

> > > ### Author Response · Authors · 2026-04-03
> > >
> > > Thank you for the follow-up. We have uploaded all code, outputs, and evaluation data to an anonymous repository and released the labeled datasets on Hugging Face as well.
> > >
> > >   Anonymous repository: https://anonymous.4open.science/r/atp-checkers-4338
> > >
> > >   Static checkers (Section 5.1):
> > >   - Checker source: https://anonymous.4open.science/r/atp-checkers-4338/src/
> > >   - Full run log across 13 benchmark variants (4,833 findings, 399 proven issues): https://anonymous.4open.science/r/atp-checkers-4338/results/fullrun.log
> > >   - Per-dataset outputs: https://anonymous.4open.science/r/atp-checkers-4338/results/
> > >
> > >   92-item semantic audit (Tables 5–6):
> > >   - Dataset + labels: https://anonymous.4open.science/r/atp-checkers-4338/annotations/semantic_lean_errors/semantic_lean_errors.json
> > >   - 6-category error taxonomy: https://anonymous.4open.science/r/atp-checkers-4338/annotations/semantic_lean_errors/error_tags.md
> > >   - Prompts and few-shot examples: https://anonymous.4open.science/r/atp-checkers-4338/annotations/semantic_lean_errors/eval_prompts_fewshot.md
> > >   -Schema: https://anonymous.4open.science/r/atp-checkers-4338/annotations/semantic_lean_errors/README.md
> > >
> > >   55-item static warning verification (Table 4):
> > >   - Dataset + labels: https://anonymous.4open.science/r/atp-checkers-4338/annotations/static_warning_verification/static_warning_verification_55.json
> > >   - Prompt template: https://anonymous.4open.science/r/atp-checkers-4338/annotations/static_warning_verification/prompt_template.md
> > >   - Provenance and schema: https://anonymous.4open.science/r/atp-checkers-4338/annotations/static_warning_verification/README.md
> > >
> > >   Hugging Face releases:
> > >   - Static warning verification: https://huggingface.co/datasets/formalanon/static-warning-verification
> > >   - Semantic audit: https://huggingface.co/datasets/formalanon/semantic-lean-errors
> > > We hope this resolves the verification concern. Please let us know if anything is missing or unclear.

---

### Official Review · Reviewer_VEGt · 2026-03-13

**Soundness:** 2
**Presentation:** 3
**Significance:** 4
**Originality:** 2
**Overall Recommendation:** 4
**Confidence:** 3

**Summary:**

The paper introduces a taxonomy for common pitfalls in formal verification benchmarks and introduces methods for detecting and avoiding each of the problems.

**Compliance With Llm Reviewing Policy:**

Affirmed.

**Final Justification:**

Very good premise and important work. I was initially on the edge about whether to recommend accepting or rejecting it because many changes would be needed (and were promised by the authors) before the camera-ready. The version they initially submitted clearly lacks the quantitative results needed for ICML. But given the importance of the work and the progress the authors appear to have made since January, I now lean towards accepting.

**Key Questions For Authors:**

Why only the small subset?

**Limitations:**

Yes

**Strengths And Weaknesses:**

*Strengths*
- This is a very important field of work, and the paper's efforts are novel.
- The taxonomy the authors present is very useful and polished, as is the list of ways to avoid them.

*Weaknesses*
The paper is a bit unsatisfying to read. The authors present all these methods to verify formal verification benchmarks, but then don't actually verify entire benchmarks. They only tried this on a small subset, about 92 problems. This subset obviously is not representative of the overall benchmark quality (and the authors don't claim it is). This means the authors do not present quantitative data on benchmark quality, nor do they have new (and improved) versions of the benchmarks in question. The claim in the abstract "We audit widely used Lean theorem-proving benchmarks" is a bit of a stretch.
This makes the paper feel like somewhere between technical documentation (explaining details of how to use Lean that are clearly not novel research) and an opinion paper. It also raises doubts about the author's methods -- if they're practical and useful to do, why not try them on the whole benchmark? (Not to mention, the small sample size affects the evaluation's soundness.) Given that there are no significant new quantitative results, I am on the edge about accepting or rejecting the paper.

---

> ### Author Rebuttal · Authors · 2026-03-31
>
> Thank you for recognizing the importance and novelty of this work, and for the direct feedback on what would make the paper more convincing.
>
> **We audited all ~10,000 problems, not 92.** The paper has two empirical components, and our submission did not include all of our quantitative results. We had run the static checkers corpus-wide but failed to include the aggregate numbers, which we correct here.
>
> Our *static checkers* (Lean 4 metaprograms) were run on every problem across all five benchmarks and their forks (13 variants). Corpus-wide results:
>
> | Benchmark | Problems | Total Findings | Proven Issues |
> |---|---|---|---|
> | FormalMath (all) | 5,560 | 3,250 | 141 |
> | ProverBench | 325 | 370 | 208 |
> | miniF2F (6 forks) | ~490 each | 74–193 per fork | 0–13 per fork |
> | ProofNet (2 versions) | 371 each | 67–82 per version | 1–5 per version |
> | CombiBench | 100 | 78 | 1 |
> | **Total** | **~10,000** | **4,833** | **399** |
>
> Total numbers across benchmarks:
> | Benchmark | Problems | Findings | Proven | Div/0 | Nat Sub | Unsound Axiom | Counterex. | Vacuous |
> |---|---|---|---|---|---|---|---|---|
> | formalmath@all | 5560 | 3250 | 141 | 1127 | 582 | 0 | 55 | 1 |
> | proverbenchmath | 325 | 370 | 208 | 81 | 7 | 199 | 5 | 0 |
> | minif2f@v2c | 244 | 193 | 8 | 45 | 28 | 0 | 3 | 0 |
> | proofnet@deepseek | 371 | 82 | 5 | 13 | 8 | 0 | 5 | 0 |
> | minif2f@yangky11-early | 244 | 79 | 13 | 14 | 19 | 0 | 5 | 1 |
> | combibench@hf | 100 | 78 | 1 | 1 | 27 | 0 | 1 | 0 |
> | minif2f@justincasher | 244 | 74 | 0 | 25 | 13 | 0 | 0 | 0 |
> | proofnet@sharp | 371 | 67 | 1 | 11 | 6 | 0 | 1 | 0 |
> | minif2f@ai-mo-test | 244 | 37 | 1 | 17 | 6 | 0 | 0 | 0 |
>
> "Proven issues" are counterexamples, unsound axioms, and vacuous theorems, for which the checker provides a certificate. The remaining findings are encoding hazards (unguarded division, Nat subtraction, analytic domain issues) requiring human or LLM-assisted triage.
>
> The *92-problem set* is separate: it evaluates whether LLMs can detect semantic errors that static analysis cannot decide. These 92 items are all problems with ground-truth error labels (from community maintenance, published corrections, manual review). There was no subsampling. This component evaluates one detection method; it is not the scope of the audit.
>
> **"Why only the small subset?"** The claim "we audit widely used Lean theorem-proving benchmarks" refers to the static checkers, which are corpus-scale. The 92-problem set evaluates one detection method, not the audit's scope.
>
> **Quantitative impact on prover scores.** We selected 20 problems with proven issues across 4 datasets, manually corrected them, and evaluated provers on both versions. Models solved 0/20 originals (flawed specifications were unprovable); on corrected versions, DeepSeek-Prover-V2-7B solved 3/20, and Kimina-Prover-8B solved 2/20. At a larger scale, Poiroux et al. independently corrected 118/371 ProofNet problems (32%), where weaker specifications make problems artificially easier. On the 186-problem test split, pass rates dropped: Claude Opus 4.5 from 22.0% to 12.4% (44% relative), DeepSeek-Prover-V2-7B from 9.7% to 6.5%, Kimina-Prover-8B from 9.7% to 7.0%.
>
> These two patterns (unprovable statements deflating scores, weakened statements inflating them) can coexist in the same benchmark and partially cancel in aggregate, making headline numbers misleading in both directions.
>
> **On the "technical documentation" concern.** Individual Lean pitfalls are documented in community resources. Our contribution shows they are *systematic across benchmarks* and provides *automated detection at scale*. The taxonomy (Table 2) classifies where faults enter the pipeline and the intervention required for each. The fidelity/loopholes/drift distinction is novel and determines which remediation applies. The empirical finding that current provers exploit flawed specifications to produce valid but meaningless proofs is a result about models, not Lean documentation.
>
> **On corrected benchmarks.** FormalMath used 12 IMO-medalist experts at 6.89\\$ per statement and achieved only 72% preservation. ProofNet had human review yet contained 32% errors. MiniF2F has undergone five independent repair efforts. If expert-reviewed datasets still show 30–50% error rates, platinum correction requires multi-annotator consensus at $300–500K across our suite, followed by ongoing maintenance. Our checkers provide the scalable first step at near-zero cost.
>
> We hope this resolves the concern about the scope. We will make the changes and add the table and other results to the camera-ready version to improve the paper's readability.

---

> > ### Author Rebuttal · Reviewer_VEGt · 2026-04-02
> >
> > Thank you for providing these valuable results. To make this work as strong as possible, I highly suggest that, for the camera-ready result you run an even larger corrected benchmark (more than just 20 problems) and include results on that.
> >
> > I'll raise my score by two points to Weak Accept.

---

### Decision · Program_Chairs · 2026-04-30

**Decision:**

Accept (regular)

**Comment:**

Reviewers agree this paper highlights an important issue and provides a useful taxonomy and audit framework. The rebuttal adds substantial quantitative evidence and resolves most concerns. Overall, a timely and practically relevant contribution.